# The Application of Nanomaterials for the Electrochemical Detection of Antibiotics: A Review

**DOI:** 10.3390/mi12030308

**Published:** 2021-03-15

**Authors:** Norah Salem Alsaiari, Khadijah Mohammedsaleh M Katubi, Fatimah Mohammed Alzahrani, Saifeldin M. Siddeeg, Mohamed A. Tahoon

**Affiliations:** 1Chemistry Department, College of Science, Princess Nourah bint Abdulrahman University, Riyadh 11671, Saudi Arabia; nsalsaiari@pnu.edu.sa; 2Department of Chemistry, College of Science, King Khalid University, P.O. Box 9004, Abha 61413, Saudi Arabia; saif.siddeeg@gmail.com; 3Chemistry and Nuclear Physics Institute, Atomic Energy Commission, P.O. Box 3001, Khartoum 11111, Sudan; 4Chemistry Department, Faculty of Science, Mansoura University, Mansoura 35516, Egypt

**Keywords:** environment, antibiotics, nanomaterials, electrochemistry, biosensors

## Abstract

Antibiotics can accumulate through food metabolism in the human body which may have a significant effect on human safety and health. It is therefore highly beneficial to establish easy and sensitive approaches for rapid assessment of antibiotic amounts. In the development of next-generation biosensors, nanomaterials (NMs) with outstanding thermal, mechanical, optical, and electrical properties have been identified as one of the most hopeful materials for opening new gates. This study discusses the latest developments in the identification of antibiotics by nanomaterial-constructed biosensors. The construction of biosensors for electrochemical signal-transducing mechanisms has been utilized in various types of nanomaterials, including quantum dots (QDs), metal-organic frameworks (MOFs), magnetic nanoparticles (NPs), metal nanomaterials, and carbon nanomaterials. To provide an outline for future study directions, the existing problems and future opportunities in this area are also included. The current review, therefore, summarizes an in-depth assessment of the nanostructured electrochemical sensing method for residues of antibiotics in different systems.

## 1. Introduction

Antibiotics are worldwide familiar as antibacterial agents and used for enhancing animal growth as well as treating diseases. Antibiotics have been widely used in medications of animals and humans due to their ability to increase feed effectiveness and therefore, increase growth rate [1,2]. Antibiotics are classified into seven branches include β-lactam antibiotics, streptogramins, lincosamides, peptide antibiotics, aminoglycosides, macrolide antibiotics, and tetracyclines [3]. Nevertheless, antibiotics have bad impacts even at trace amounts on human health resulted from their ability to accumulate inside food chains and causes many dangerous symptoms such as organ toxicity, and hearing loss [4]. Furthermore, diseases treatments efficiency decreases by antibiotics abuse as a result of the associated increase of resistance genes occurrence [5]. Additionally, animals and humans could be exposed to the real danger represented in the growth of antibiotic-resistant bacteria that are expected to develop to other microbial populations [6]. Thence, human safety and health particularly required the control of antibiotics.

The quantitative/qualitative detection of antibiotics via different analytical techniques was the aim of specialists in the last few years. They spent efforts and time developing accurate and fast detection methods. The most antibiotics familiar determination methods are high-performance liquid chromatography (HPLC) [7], gas chromatography combined with electron capture (GC-EC), gas chromatography coupled mass spectroscopy (GC-MS) [8], and thin-layer chromatography (TLC) [9]. But, many obstacles prevent the application of chromatography as a tool for antibiotics detection such as expensive time and apparatus. There are many other accurate and precise antibiotics detection methods such as enzyme-linked immunosorbent assay (ELISA), flame ionization (FI), diode array (DA), and capillary electrophoresis (CE), however, these techniques require highly trained technicians and complex samples preparation that hinders their wide application [10]. Consequently, human safety and health required the development of rapid, robust, convenient, selective, and sensitive techniques for the determination of antibiotic residues. The electrochemical biosensors have several advantages such as in-situ applications, fast detection, and great selectivity that enhance their application for the analytical detection of antibiotics [11].

Nowadays, analytical techniques used for drug analysis, food analysis, environment, and health include nanomaterials as the main constituent as nanomaterials can improve the interaction with analyte due to their exceptional electronic, chemical, physical properties as well as their high surface area [12,13,14,15,16,17,18,19,20]. Several factors affect the efficiency of nanomaterials-based sensors such as dimensions, morphology, crystallographic axis orientation, crystal structure quality, and chemical composition that determine the rate of electron transfer and degree of interaction [21]. Thus, clinical and analytical instruments used nanomaterials for sensing purposes due to their exceptional potential. So, all nowadays sensors with different principles (electrochemical impedance, potentiometric, and amperometric) were improved to proper their applications. However, the existence of nanomaterials in the electrochemical sensor allows the signal maximization and selectivity to the certain analyte; therefore, determines sensor performance [22]. Many previous works discussed the application of nanomaterials for electrochemical sensing of antibiotics [23,24]. We aim in this review to update and organize the nanomaterials applied for the electrochemical sensing of antibiotics. Until now, antibiotics electrochemical sensors discussed how nanomaterials can be utilized to improve their performance toward antibiotics in different systems. This review addresses challenges faced by the application of nanomaterials in antibiotic sensors and recommendations to improve these sensors in the future. Consequently, this review will be a reference to scientists’ aim to study the nanomaterials-based antibiotic sensors to start where the other ends. 

## 2. Antibiotics Electrochemical Detection Methods

Our aim in the next section is the discussion of voltammetric sensors used for the electrochemical detection of antibiotics with the discussion of the factors affecting their performance. Recently, the antibiotic residues were detected accurately via electrochemical sensors that have several advantages over other techniques such as miniaturization possibility, fast time of analysis, and exceptional sensitivity resulted from the coupling between time (t), charge (Q), current (I), and potential (E) [25,26]. Until now, the most frequent technique used for the electrochemical detection of antibiotics is voltammetry with its all types cyclic voltammetry (CV), differential pulse voltammetry (DPV), square wave voltammetry (SWV), and stripping voltammetry (SV). In general, voltammetric sensors have been attracted attention due to their improved performance in electroanalysis. Additionally, amplification of mass-transport, simple measurement and preparation, the possibility of miniaturization, and automation are the well-known advantages of voltammetric techniques. The fast response, selectivity, low detection limits, and wide linear range are additional advantages of such sensors. CV is an electrochemical method in which the voltage is too high and the corresponding developed current is measuring. The resulting current is measuring and the working electrode potential is cycling. The typical CV electrochemical cell is consisting of a reference electrode that maintains the constant potential, a counter electrode that closes that circuit between the working electrode and signal source, a working electrode at which the reaction takes place, and an electrolytic solution that provides ions necessary for a redox reaction. This cell is connecting to a potentiostat that produces the accurate potential and allowing the small current signal drawing. The resulting current is measuring using a current to voltage converter while the final voltammogram is producing by the data acquisition system. DPV is an electrochemical method in which the application of amplitude potential pulses to a linear ramp potential is involving. The electrode base potential value is choosing at the value corresponding to the faradaic reaction absence. In between pulses with identical increments, the base potential increases. At the pulse termination and before the pulse application, the current is measuring immediately with the recording of the difference between them. The resulting pulse current voltammograms have a differential shape. SWV is considered one of the most sensitive and fastest pulse voltammetry methods available with comparable detection limits to that of spectroscopic and chromatographic methods. Mechanism and kinetics of the electrode process under investigation can evaluate using this technique by the analysis of its characteristic parameters. The potential current curve shape is resulting from the variation of the period and the potential step that produces the potentials of height Δ*E*. SWV measurements occur before an initial time (ti) when the working electrode is polarized at a potential where no redox reaction occurs. This technique offers high rejection to capacitive currents and excellent sensitivity. One of the main families of electroanalytical chemistry is the SV technique. The low detection limits accompanying the SV technique are the source of this technique’s strength. A pre-concentration step involving the accumulation of the analyte metal over the working electrode surface is responsible for this technique’s excellent sensitivity. After the preconcentration, the preconcentrated analyte is stripped back to the solution and this step is the measurement (stripping) step in which there is a direct proportional between the analyte concentration in the sample and the current response. The detection of antibiotics using electrochemical impedance spectroscopy and chronoamperometry were considered as parallel options. Details of these methods including the limit of detection, application, and principles were described in Table 1.

### 2.1. Antibiotics Electrochemical Detection Strategies

Several electrochemical strategies were used for the electrochemical detection of antibiotics. The first concept depends on the synthesis of molecularly imprinted polymers (MIPs) with definite cavities on their surface corresponding to the target analyte for selective detection. The second concept is depending on the use of aptamers for selective and sensitive detection of the analyte by including DNA and RNA in the sensor’s components. The other concept is depending on the complex formation between antigen and antibody (immuno-complex). The last concept is depending on enzyme/receptor linkage. Nanomaterials can be used according to these strategies for the detection of antibiotics. The next sections discuss the strategies used for antibiotics sensing.

#### 2.1.1. Biosensors Based on Molecular Imprinted Polymers (MIPs)

The existence of analyte definite receptors over molecular imprinted polymers-based sensors attracts interest recently. MIPs synthesis occurs via a reaction mixture contains a polymerization initiator, a cross-linking monomer, a functional monomer, and a template. A three-dimensional polymer network is formed due to the complex formation between functional monomer and template that is surrounded by cross-linking monomer. After polymerization ending, templates are removed by washing leaving cavities behind [26]. The template is selected according to the target analyte. MIPs-based sensors for the detection of antibiotics with high sensitivity and selectivity are coupled with several nanomaterials like graphene, metal nanoparticles, magnetic nanomaterials, Prussian blue catalytic polymer, and graphene oxide [27,28,29,30,31,32].

#### 2.1.2. Biosensors Based on Aptamers (Apts)

Antibiotics detection via biosensors based on aptamer (RNA or single-stranded DNA molecules) attracted attention as a powerful tool recently. Biosensors using aptamers as recognition components are known as aptasensors. The aptamer molecule of the aptasensor binds the target analyte with high specificity and affinity. Biosensors based on RNA aptamers are usually only used for one-shot detections because they are quickly degraded by endoribonucleases found in the biological environment. While Biosensors based on DNA aptamers are generally stable and can be synthesized with high reproducibility and purity, making biosensor fabrication processes simpler. Simple modification procedures of aptamers, as well as simple in-vitro production, low cost, long shelf life, and good physicochemical stability, are advantages that distinguish aptamers [33]. Two main techniques are used for the immobilization of aptasensors. Firstly, functionally activated surfaces are non-covalent modified with aptamers. In this method, the strong affinity between two molecules is exploiting for the immobilization of aptamer over the sensor surface. The sensor surface is modifying with one molecule and the aptamer with the other that allows the non-covalent (physisorption) immobilization of the aptamer over the sensor surface. An example that is used in many biosensors is the efficient immobilization of biotin to avidin in which avidin-coated sensor is incubated with biotin-tethered aptamer in buffer at room temperature for the physisorption immobilization. Secondly, appropriate linkers are used for direct modification of a bio-functionalized sensor surface with aptamers. In this method, linker groups (such as biotin, amine, and thiol) are connected to the terminal part of the aptamer during its fabrication to allow the connection of aptamers with sensor surfaces to enhance the accessibility of studied analytes to the aptamer binding site. During this method, the position of functional groups can be accurately controlled which allows the highly controlled orientation of aptamers during immobilization providing high binding activity more than immobilization in the random orientation that led to the loss of functionality. The sensitivity and selectivity of aptasensors can be enhanced for the electrochemical sensing of antibiotics by coupling with familiar nanomaterials like palladium nanoparticles, gold nanoparticles, and carbon fibers [34,35]. Moreover, heavy metal ions doped metal-organic frameworks (MOFs), ferrocene, and methylene blue (MB) were used to improve the sensitivity of aptasensors and produce characteristic signals for multiplexed antibiotics [36,37,38]. 

#### 2.1.3. Biosensors Based on Immuno-Complex

The electrochemical immunosensor, a form of the biosensor, uses the antibody as a capturing tool and tests the electrical signal generated by the antibody attaching to the target molecule quantitatively. Figure 1 shows the general principle of electrochemical immunosensor fabrication. According to Figure 1, the complex reaction between analyte and antibody is the source of electrochemical signals. The resulted product affects the electric charge that can further be detected by the electrode. Several advantages were achieved by using immunosensors depend on antigen-antibody reaction such as high sensitivity, cost-effectiveness, and high reliability toward analytes such as food veterinary residues, environmental contaminants, and drugs [39]. But, specific “antigen-antibody” interactions required the incorporation of electron mediators into the detection process [40]. Additionally, the sensitivity of immunosensors toward low antibiotics concentrations is generally very low. So, biomolecules immobilization efficiently over the surface of electrode required the efforts to fabricate unique nanomaterials. Several nanomaterials are potentially used to achieve antibiotic label-free immunosensing such as magnetic nanomaterials [41], polymers [42], nanoporous gold [40], and carbonaceous nanomaterials [43]. Regardless, antigen-antibody complexes can be incorporated with dyes [44], quantum dots [45], and enzyme labels [46] to enhance their sensitivity toward antibiotics.

#### 2.1.4. Enzyme/Receptor-Mediated Biosensors

Several types of antibiotics can be detected accurately via nanomaterials labeled many receptors/enzymes. Most of these studies state several nanomaterials such as magnetic nanoparticles, carbonaceous nanomaterials, and colloidal gold tags were incorporated with different enzymes like glucose oxidase, penicillinase, and horseradish peroxidase for the electrochemical sensing of antibiotics [48,49,50].

### 2.2. Antibiotics Electrochemical Sensing Using Nanomaterials

Nanomaterials are the greatest attractive choice for the fabrication of functional electrochemical biosensors due to their exceptional electronic, chemical, optical, thermal, and physical properties. The different ways by which the nanomaterials are synthesized as shown in Figure 2 provides variety in their size, shape, and physical and chemical properties that allow their application in different fields. The ease of functionalization and modification of such nanomaterials-based biosensors facilitated the improvement of their efficiency toward the detection of antibiotic residues. In this section, the different types of nanomaterials applied for the electrochemical sensing of antibiotics are discussed.

#### 2.2.1. Quantum Dots

Quantum dots (QDs) have many properties such as ease surface modification, fast electron transfer, acceptable interfacial properties, brilliant biocompatibility, and great aspect ratio that attract attention for use in different bio-sensing applications [52,53]. QDs are signal tags in the multiplexed biosensing processes via utilizing the oxidation potentials of the accompanying metal ions [54]. However QDs excellent properties, agglomeration possibility, hard and long synthesis procedures, and toxicity of heavy metals QDs may hinder their biosensing applications [55]. So, the latest developments of antibiotics QDs based sensors were summarized in this section. 

Ciprofloxacin facile electrochemical sensing was achieved via complex formed between ciprofloxacin and in situ liberated Cd^+2^ ions from CdS quantum dots [56]. Excellent ciprofloxacin detectability was achieved by this sensor that was found to give a detection limit of 22 nM with high selectivity toward other interfering antibiotics (gentamicin, erythromycin, kanamycin, chloramphenicol, and ofloxacin). The recovery of any sensor is a critical element in comprehending the material’s application in large-scale real samples. The spiking method is used for the determination of the recovery of developed electrochemical sensors in which the test samples were provided with known amounts of the analyte. Un-spiked test samples are analyzed parallel to spiked test samples. So, the recovery percentage may be less or more than 100% depending on the recovered amount of analyte compared to the identified amount added. This sensor provides a good ciprofloxacin recovery range in real samples (human urine) from 99.2% to 109.7% indicating the validity of this sensor for ciprofloxacin detection in real biological samples. In another study, to avoid the CdS QDs toxicity a composite of graphene quantum dots (GQDs) and poly(o-aminophenol) synthesized via electropolymerization for the voltammetric detection of levofloxacin [57] with a very low detection limit of 10 nM reached due to fast electron transfer and excellent electrical conductivity of GQDs. Similar levofloxacin structures include norfloxacin, lomefloxacin, enrofloxacin, and ciprofloxacin were used to test the sensor response that provides excellent recovery of 97% to 102% from real samples (spiked milk). Chen et al. exploited QDs potential with multiplexed assays of antibiotics to fabricate low-cost devices [37]. They detected chloramphenicol antibiotic in the attendance of polychlorinated bi-phenyl-72 organic contaminant via voltammetric assay based on magnetic aptamer–modified QDs encoded dendritic nanotracer. The resulted detection limit toward chloramphenicol was 1.0 pM reached due to dendritic nanotracers amplification properties. This value is very low compared to commercially used kits of enzyme-linked immunosorbent assay (ELISA) that approve the high sensitivity of the sensor. The sensor showed high specificity beside sensitivity toward like structures such as oxytetracycline and kanamycin. Additionally, this sensor provided suitable antibiotics recovery from real samples (environment and food) ranging from 93% to 102.2%. Moreover, QDs-based tags and aptamers provided a very sensitive determination method for tetracycline, chloramphenicol, and streptomycin antibiotics combined with a too low limit of detection equal 20 nM for tetracycline, 5 nM for chloramphenicol, and 10 nM for streptomycin in milk [23]. These very low values were achieved due to the optimal synergy. Additionally, the highest response of sensor toward studied antibiotics achieved via discovering the impact of different factors includes antibiotics incubation period and concentrations of different DNAs that used for QDs modification.

#### 2.2.2. Metal-Organic Frameworks

Among highly porous materials, Metal-organic frameworks (MOFs) polymers have high specific surface areas, well-defined pore-sizes, and large internal pore volumes [58]. Moreover, the prolonged crystalline structures of MOFs in which metal ions are combined using linker organic molecules [59] as shown in Figure 3 allowed different sensing applications due to many options for the modification of organic linkers and metal ions as well as good tunability of MOFs in size of pores from nanometers to angstroms. 

However, all advantages of using MOFs in the fabrication of the sensors, many limitations are associated with MOFs-based electrochemical sensors such as long response time, deficiencies in target selectivity, poor binding of targets to MOF sensing surface, and the difficulties of the synthesis of conductive MOFs as MOFs are semiconductors or insulators. All these limitations can overcome through a post-synthetic chemical modification that allows the fabrication of MOFs-based sensors. In the next section, we discussed the application of MOFs toward the potential detection of different antibiotics. For the first time, tetracycline in honey samples was detected using molecularly imprinted polymer (MIP) microporous MOFs [61]. In this study, the gold (Au) electrode surface was modified using imprinted composite via electro-polymerization of imprinted molecules with p-amino-thiophenol functionalized Au NPs. To achieve the highest electrochemical response toward tetracycline antibiotics, the authors studied the impact of different factors such as elution conditions, washing conditions, incubation time, and extraction time. This sensor showed a detection limit of 0.23 fM toward tetracycline in a range of concentration equal 225 fM to 22.5 nM. The existences of similar structures such as doxycycline are not affected on the response of the sensor that also, provided acceptable recovery (102–107%) for detecting tetracycline in samples of honey. Thus, this sensor appears to be proper for the monitoring of tetracycline in the environment and food samples even at very low concentrations. Similarly, chloramphenicol antibiotic was detected using chromium-containing MOF [62]. Chromium-containing MOF has very low conductivity that made the author’s mix chromium-containing MOF with high conductive carbon black XC-72 to produce a unique hybrid composite with a detection limit of 1.6 nM toward chloramphenicol in the range of concentration equal (0.01 to 21 μM). Additionally, the authors achieved a maximum response of the sensor toward the antibiotic by studying the impact of different factors such as pH, scan rate, accumulation time, and the ratio of chromium-containing MOF: carbon black XC-72. The response of the sensor toward chloramphenicol was found to be higher than that of a glassy carbon electrode (GCE) modified separately by carbon black XC-72 or chromium-containing MOF. The fabricated sensor was examined for response toward several interfering antibiotics such as amikacin, gentamycin, neomycin, penicillin, kanamycin, kiasamycin, tetracycline, and chlortetracycline. This sensor improved response toward the detection of chloramphenicol antibiotic resulted from the combination between carbon black XC-72 great conductivity and chromium-containing MOF high surface area. This fabricated sensor is considered to be very proper for real sample analysis like eye drop, honey, and milk. Chen et al. extend the use of MOFs for multiplexed determination of kanamycin and oxytetracycline antibiotics [36]. Here, an ultrasensitive aptasensor was fabricated using signal tracer from metal ion-doped MOFs and RecJ_f_ exonuclease-catalyzed targets recycling amplification. The aptasensor is composed of (nano-sized MOFs labeled with metal ions and aptamers) and of capture beads (anti-single-stranded DNA (anti-ssDNA antibody) carried on Dynabeads). Mainly, the amino groups of MOFs (UiO-66-NH_2_) abundant inside ultrahigh pores were used to carry metal ions of cadmium and lead that then label aptamers specified toward antibiotics of kanamycin and oxytetracycline. So, the sensor here is fabricated depending on specific complexation between aptamers and anti-ssDNA antibodies. Aptamers favor the formation of targets-Apts-MNM instead of an anti-ssDNA antibody-aptamer that consists due to the liberation of Apts-MNM from capture beads. Moreover, target antibiotics were dissociated from RecJ_f_ exonuclease to create higher signal tracers detected via square wave voltammetry with ultrasensitive detection toward kanamycin and oxytetracycline equal to 0.15 pM and 0.19 pM, respectively. The sensor provided accurate detection even in the presence of interfering antibiotics like gentamicin sulfate, streptomycin, chlortetracycline, chloramphenicol, and doxycycline that easing the application for analysis of real samples like milk. The same group later fabricated multiplexed sensor toward chloramphenicol and kanamycin antibiotics [63]. This novel sensor was fabricated by label metal ions of Cd (II) and Pb (II) on nano-sized MOFs through binding the pores amino groups (UiO-66-NH_2_). These bio-codes save adsorption capacity toward Cd (II) and Pb (II) equal 268.4 and 293.5 mg/g, respectively for signal amplification. Then, complementary DNA strands of aptamers toward two antibiotics were used to label metal-nanoMOFs that were later combined with aptamer functionalized magnetic beads to contain the bio-codes specific toward two antibiotics. The presence of magnetic beads allowed the rapid release of Metal-nanoMOFs due to the conjugation between antibiotics and aptasensor that improve the detection process using the square wave voltammetry transduction method. Within a concentration range of 0.7 to 50.0 μM of two antibiotics, the bio-codes-based sensor provided ultra-sensitivity with a detection limit of 0.17 pM and 0.18 pM for kanamycin and chloramphenicol, respectively. Additionally, this excellent electrochemical sensor was potentially used for the detection of kanamycin and chloramphenicol in real samples of milk. The sensor efficiency was not affected when tested for interfering structures such as Mg (II), Ca (II), K (I), chlortetracycline, and oxytetracycline). This group then fabricated an electrochemical aptasensor using a Y-shaped DNA probe for the determination of multiplex antibiotics [64]. The nano MOFs encoded with probes-based metal ions act as a substrate, then the amplification strategy is initiated via circular strand replacement DNA polymerization. Another assisted DNA probe labeled with Au magnetic NPs used to assemble Y-shaped DNA probes. Thus, Y-DNA probes can be hybridized with the captured DNA probe (composed of primer recognition region and aptamer) and signal tags (several metal ions ex. Cd (II) or Pb (II) and nano MOFs encapsulating DNAs). The excellent electrochemical signals were produced due to the efficient packing of many metal ions on the large surface area saved by the nano-MOFs substrate. Furthermore, the attraction between target antibiotics (oxytetracycline and chloramphenicol) and aptamer domain (more than with complementary sequence) triggered the conformational change that can release the signal tags that then via the Bst DNA polymerase can be amplified by cyclic-induced polymerization. The sensor provided a detection limit of 34 fM for chloramphenicol and 49 fM for oxytetracycline meaning that the electrochemical signals were folded by 17-fold. Additionally, this fabricated sensor provided unaffected selectivity to the two target antibiotics even in the presence of interfering structures that include antibiotics (gentamicin sulfate, tetracycline, kanamycin, and doxycycline), metal ions (Fe (III), Ca (II), K (I), and Mg (II)), and proteins (β-lactoglobulin, casein, and α-lactalbumin) that usually exist in milk. The replacement of appropriate aptamers and the modification of metal ions for the probe allow the use of this assay universally for food safety analysis. Similarly, a new very sensitive electrochemical sensor based on exonuclease and endonuclease to provide a dual recycling amplification strategy was fabricated for the multiplex detection of kanamycin and chloramphenicol [38]. Over a wide concentration range (10^−4^ to 50 nM) for both antibiotics, there was a linear correspondence between the concentration and current intensity during the analysis using the square wave voltammetry transduction method. Signal tags were potentially encapsulated using embedded HP-UIO-66-NH_2_ but, Bam HI and EcoRI restriction endonucleases were amplified and enriched via Au NPs. Several signal tags were liberated into the reaction medium due to the presence of target antibiotics led to the signal amplification up to 54 folds with detection limits of 22 and 36 fM toward chloramphenicol and kanamycin, respectively. The sensor sensitivity was not affected toward targets in the presence of interfering structures include erythromycin, oxytetracycline, thiamphenicol, tetracycline, and florfenicol that enhance the use of fabricated sensor toward the detection of target antibiotics in real samples as fish samples. Compared to ELISA commercial kits, this sensor provided results that promote its application in detecting food samples with multiplexed antibiotics. Lately, Ce-based metal-organic framework (Ce-MOF) and porous organic framework (COF) were used to fabricate nanohybrid of (Ce-MOF@COF) as a label-free biosensor for the detection of antibiotic oxytetracycline [65]. Over the antibiotic concentration range of 0.3–1.0 nM, the fabricated sensor provided a linear electrochemical response with a 36 fM detection limit. High bio-affinity toward oxytetracycline-specific aptamers, the ability of aqueous dispersion, and unique electrochemical performance are the reasons for improved performance of the nanohybrid sensor. Appropriate practical applicability, stability, high selectivity, and great reproducibility of the aptasensor were combined with exceptional performance for the detection of oxytetracycline in real samples including urine, wastewater, and milk.

#### 2.2.3. Metal Nanoparticles

Metal nanoparticles are a class of materials that have structure, shape, composition, and size responsible for their good chemical and physical properties. Their excellent conductivity, structural robustness, mechanical and electronic properties, biocompatibility, and catalytic made metal NPs the most common candidates for electrochemical sensors-based NPs. Recently, huge development has been achieved in the fabrication of novel metal nanostructures and their effective application in different fields like medicine, sensors, electronics, and catalysis. Metal nanoparticles have attracted considerations for the application in the electrochemical detection of antibiotics due to numerous adsorption sites to antigen, enzymes, and antibodies, fast electron transfer kinetics, and very high surface area [66]. Thus, metal nanoparticles provide the proper substrate for the immobilization of enzymes, antibodies, proteins, and aptamers. Consequently, the recent improvements in the application of metal nanoparticles for the electrochemical detection of antibiotics were discussed in the next section of the review. Gold (Au) nanoparticles have unique and promising physicochemical properties such as electrocatalysis features, accepted biocompatibility, high surface area, and high conductivity which allow their wide sensing application in biological and chemical samples. From all-metal NPs, gold NPs have been extensively studied for enhancing electrochemical biosensors detection limits. Moreover, antibiotics detections using different electrochemical sensing strategies were achieved by the combination of bio-recognition molecules (antibodies, enzymes, and aptamers) with the unique sensitivity of Au NPs. In this context, daunomycin was selectively and sensitively detected using a label-free electrochemical aptasensing method [67]. In this sensor, Au NPs functionalized with a conducting polymer such as 2, 2′:5′2″-terthiophene-3′ (p-benzoic acid) used for effective co-immobilization of aptamers and phosphatidylserine. Authors tested the effect of different factors to get the maximal sensitivity such as reaction time, temperature, immobilized aptamer concentration, and pH which provided a detection limit of 0.053 nM over a concentration range of 0.1 to 61 nM toward daunomycin antibiotic. The fabricated sensor provided brilliant selectivity, sensitivity, and stability toward daunomycin detection in real samples of human urine. Similarly, kanamycin was detected using another fabricated label-free electrochemical aptasensor constructed by covalent immobilization of kanamycin specific DNA aptamer over Au NPs previously modified with poly–[2,5-di(2-thienyl)-1H-pyrrole-1- (p-benzoic acid)] [68]. Using (linear sweep voltammetry) LSV as a transduction method, this fabricated aptasensor provided over a concentration range of 0.06 to 10 μM a detection limit of 9 nM. Additionally, the self-assembled poly–[2,5-di(2-thienyl)-1H-pyrrole-1- (p-benzoic acid)]/Au NPs based detecting strategy provided excellent sensitivity of the aptasensor due to eased electron transfer for a signal transducer that reached 4.3 × 10^−3^ μA μM^−1^ cm^−2^. Kanamycin antibiotic sensing was further advanced through electrochemical ultrasensitive enzyme-free detection based on gold NPs biomimetic peroxidase activity with the target-induced replacement of a kanamycin aptamer [69]. The sensor was found very proper for the application to kanamycin detection in environmental and food samples due to its unaffected sensitivity by interferes antibiotics such as (tetracycline, neomycin, streptomycin, and gentamycin) as well as its very low detection limit of 0.07 nM over a concentration range of 0.1 to 61 nM using differential pulse voltammetry (DPV). Furthermore, ultra-sensing of metronidazole has been achieved using a new strategy of the support-less electrochemical platform via applying molecularly imprinted polymer [70]. In this study, metronidazole was simultaneously quantified and identified using molecularly imprinted polymer prepared by electropolymerization and modified using porous Au–Ag micro-rods. This sensor provided a very low detection limit toward metronidazole of 0.03 pM due to the association between the molecularly imprinted polymer and nanoporous free-standing microelectrodes. Additionally, besides the good selectivity toward metronidazole, the sensor showed selectivity toward analytes such as NO_3_^−^, Cl^−^, K^+^, Na^+^, NH_4_^+^, and Ca^+2^. Tetracycline was detected using a novel label-free electrochemical sensing concept [71]. The anti-tetracycline aptamer was efficiently immobilized via electrodepositing methylene blue and gold NPs over a glassy carbon electrode (Figure 4a) that allows the formation of Au–S bonds. The maximized voltammetric response of fabricated electrode (Figure 4b) achieved due to tetracycline maximized adsorption by methylene blue, improved electron transfer kinetics of Au NPs, and well-dispersed nanoparticles over a glassy carbon electrode without any agglomerations (Figure 4c). 

Subsequently, the fabricated electrode showed a detection limit toward tetracycline of 4 pM with linear detection of 1 × 10^−4^ to 1 × 10^3^ μM. Additionally, the sensor was tested for selectivity against interferes like kanamycin monosulfate, gentamycin sulfate, and oxytetracycline (Figure 4d), reproducibility, and stability when detected milk samples tetracycline. In another study, a label-free electrochemical aptasensor was fabricated using polyhedral-shaped Au nanocubes that were prepared via the seed-mediated method to act as electrochemical detective amplifiers for the detection of chloramphenicol antibiotic [72]. Selective chloramphenicol detection was achieved using a screen-printed electrode labeled with aptamer/AuNCs-cysteine. The aptasensor provided good sensitivity toward chloramphenicol detection with linear ranges of (0.26–6.0 and 0.04–0.10 μM) and detection limits of 5 nM. Additionally, the sensor was found applicably for chloramphenicol detection in human serum with excellent stability and selectivity. After that work, chloramphenicol was detected via an alternative cost-effective aptasensor fabricated using dendritic Au nanostructures [73]. The fabricated aptasensor contained a screen-printed graphite electrode coated with mesoporous silica-supported with 1,4- diazabicyclo[2.2.2]octane, molecular recognition element, chloramphenicol binding aptamer, and dendritic Au nanostructures. The sensor showed detection limits of 5 nM toward chloramphenicol detection with two linear ranges (0.16–7.0 and 0.04–0.15 μM) using DPV as a detection method. The sensor was found to be promising for chloramphenicol sensing in human serum with improved repeatability, selectivity, stability, and sensitivity. This research group detected chloramphenicol using label-free electrochemical aptasensors based on Au nanostructures in two successive works [72,73]. The same group after that was detected epirubicin selectively using label-free electrochemical aptasensor [74]. In this sensor, the Au-S bond caused the attraction between magnetic double-charged diazoniabicyclo [2.2.2] octane dichloro silica and electrodeposited Au NPs that were used to modify carbon screen-printed electrode coated with self-assembled 5′-thiole terminated aptamer (exact to epirubicin) to get an ultra-low detection limit of 0.05 μM toward epirubicin. This sensor was promising for the detection of epirubicin in human serum due to excellent selectivity (Imatinib, Tamoxifen, Docetaxel, and Paclitaxel) and sensitivity with two detection ranges (2.0–21.0; 0.08–1.0 μM). As well as aptamers application successfully for antibiotics detection, enzymes have been grouped with Au NPs to improve the sensor selectivity and sensitivity [75]. In this context, sulfamethoxazole was detected using an amperometric biosensor fabricated by the modification of the screen-printed electrode with a tyrosinase crosslinked Au nanoparticle. The sulfamethoxazole was detected over the fabricated electrode with a detection limit of 23.0 μM and a linear range of 0.03–0.2 mM. Recently, kanamycin was detected using an electrochemical aptasensor fabricated by functionalizing Au NPs with horseradish peroxidase to reach the signal amplification [76]. Trace concentration of kanamycin was detected due to NPs accompanied signal amplification and enzyme-based kinetic reactions. The signal amplification of the NPs and horseradish peroxidase accompanied signal amplification produced an ultra-low detection limit of 2 pM over a detection range of 4 × 10^−5^ to 0.2 μM. Kanamycin was approved to be detected using this aptasensor in milk real samples due to its unaffected selectivity in the presence of interferences include gentamycin, streptomycin, florfenicol, thiamphenicol, and chloramphenicol). Moreover, brilliant catalytic efficiencies and the large surface area of different metal NPs enhanced their application for the electrocatalytic detection of antibiotics. Thus, cefixime was electrochemically detected over multi-walled carbon paste electrode modified Au NPs [77]. Cefixime maximal electrochemical response was achieved through studying the effect of different factors on the voltammetric wave-like time, scan rate, accumulation potential, and pH. Cefixime adsorption was controlled at electrode interfaces with an irreversible oxidation process. The electrode showed excellent repeatability, reproducibility, and sensitivity over than results in literature with very low detection limits of 4 nM over a concentration range of 0.02 to 200 μM due to shared advantages of both Au NPs and carbon paste electrode. The existence of interferences like Mg^+2^, NO_3_, Ca^+2^, K^+^, NH_4_^+^, caffeine, glucose, oxalic acid, citric acid, ascorbic acid, and uric acid not affected the performance of the sensor. Similarly, tetracycline was determined by direct electrochemical concept over novel sensor fabricated by the electrodeposition of Au colloids over tungsten tip of microelectrodes [78]. The fabricated sensor provided low detection limits of 0.3 μM over a concentration range of 2.5 to 22 μM toward tetracycline detection with a linear dependence of currents oxidation peaks on antibiotic concentration. An ultrasensitive ceftizoxime imprinted electrochemical sensor was fabricated using composites of 5-(5-bromo-2-hydroxybenzylidenamino)-2-mercaptobenzimidazole IL and Ag@Au NPs [79]. The fabricated sensor showed a very low detection limit of 3.0 pM over a detection range of 1.0 to 0.02 nM with excellent stability, repeatability, and specificity toward the detection of ceftizoxime antibiotic. In another study, carboxymethyl cellulose stabilized magnetite nanostructures were modified using gold NPs to fabricate an electrochemical amplification platform for the sensing of chloramphenicol antibiotic [80]. The sensor showed a very low detection limit of 0.07 μM over the detection range of 2.6 μM to 26 μM with excellent linearity toward chloramphenicol detection using the SWV method. The sensor was found to applicable for chloramphenicol sensing in human urine due to resulted recovery value of 98%. Other than Au nanoparticles, tetracycline antibiotics were detected using Pt NPs supported with carbon [81]. To obtain the maximal electro-oxidation response of tetracycline, the authors studied the effect of different factors like pH, electroactive surface area, scan rate, and amount of Pt NPs. The sensor showed excellent reproducibility and redox behavior toward tetracycline detection due to the excellent combination of stability and activity of carbon/Pt NPs. The composite showed a low detection limit of 4.30 μM over the concentration range of 10 to 44.00 μM toward tetracycline detection. The authors approved the effective application of the fabricated sensor in clinical studies due to its effectiveness in the detection of low concentrations of tetracycline in urine samples. Similarly, ceftriaxone was detected in the presence of lidocaine using a voltammetric sensor fabricated using multi-walled carbon nanotubes (MWCNTs) and Pt NPs [82]. The multi-walled carbon nanotubes and Pt NPs synergistic effects allowed the low detection limit of 9.00 nM toward ceftriaxone over the concentration range of 0.02 to 10 μM.

#### 2.2.4. Magnetic Nanomaterials

Recently, attentions have been increased toward the improvement and application of magnetic nanoparticles in biosensors due to their properties like easy synthesis, biocompatibility with biomolecules, provide immobilization substrates for bio-recognition residues (peptides, aptamers, and antibodies), exceptional physicochemical properties, high mass transference, high surface area [83]. The simple synthesis, modification, and functionalization of magnetic nanoparticles were presented in Figure 5. Magnetic nanoparticles have a quick response in magnetic fields that make them proper in different applications especially at a size below 50 nm produce superparamagnetism and achieve the best at 10–20 nm [84].

Magnetic nanomaterials have surface functionalities that enable them to be modified with biorecognition elements that have a high affinity for the analyte of interest, and the broad surface-to-volume ratio increases the number of bioreceptors available to react with the analyte, thus increasing the device’s sensitivity. Additionally, Magnetic nanoparticles are appropriate to chelate analytes in the test by applying an external magnetic field and to be added into the transducer materials [86]. Importantly, besides all previous advantages of magnetic nanomaterials, immunosensors based on the magnetic labels used the magnetic separation of biomolecules from complex biological samples and as solid supports for performing immunoassays. This magnetic separation of the analyte will reduce the number of false positives by minimizing matrix results. Further, in several immunosensors based on magnetic nanomaterials, there is an application of an external magnetic field during the detection for more improved results to force the complexation reaction between the analyte and magnetic nanomaterials. The application of a magnetic field force to drag the analyte-MNP complex against the sensor surface lowers the limit of detection (LOD) by 2400 times compared to the LOD achieved in the absence of a magnetic field. So, magnetic NPs based sensors have many properties such as quicker analysis, less noise, lower detection limit, and improved sensitivity compared to non-magnetic-based sensors [87]. According to several properties of magnetic NPs, different electroanalytical methods have been explored for the electrochemical sensing of antibiotics residues in diverse samples like beverages, food, environmental, and human fluids samples.

Firstly, electrochemical immunosensors for different antibiotics were fabricated via a combination between magnetic NPs and antibodies. Kanamycin antibiotic was sensed using label-free electrochemical immunosensor that fabricated as nanocomposite of thionine (TH) functionalized graphene sheets (GS), Ag NPs, and magnetic NPs [88]. In this study, graphite powder was used to synthesize graphene oxide via modified Hummer’s method followed by the reduction of graphene oxide to produce graphene sheets. Then the thionine as a mediator was adsorbed on the graphene sheets by π–π stacking to produce TH-GS that was used to modify the GCE surface. After that, the synthesized Ag@Fe_3_O_4_ NPs were mixed with kanamycin antibodies for 1 day to produce Ag@Fe_3_O_4_ –Ab. The glutaraldehyde (GA) solution as a bifunctional linker was added to the electrode surface to save aldehyde groups for the combination between TH-GS and Ag@Fe_3_O_4_ –Ab. Finally, an electrode modified by TH-GS/GA/Ag@Fe_3_O_4_–Ab as a label-free immunosensor was used for kanamycin detection. Each component of this TH-GS/GA/Ag@Fe_3_O_4_–Ab label-free immunosensor has a role in the sensitive detection of kanamycin. Fe_3_O_4_ NPs can enhance the electron transfer and immobilize more kanamycin antibodies while the addition of Ag NPs to form hybridized NPs (Ag@Fe_3_O_4_) can further enhance the electron transfer efficacy by keeping the advantages of both Fe_3_O_4_ and Ag. Ag@Fe_3_O_4_ NPs large surface area could improve the immobilizing quantity of antibodies. GC enhanced the electron transfer and provided the large specific surface area that causes the sensitivity amplification. The peak current change of TH before and after the antigen-antibody reaction was used for the sensitive detection of kanamycin as the complex formed between the antigen (antibiotic) and antibody can hinder the electron transfer through the electrochemical sensor. Thus, the sensitivity toward the antibiotic has been enhanced in two ways, enlarging responding electrochemical signals and increasing loading capacities of antibodies. In this study, kanamycin was accurately detected over the concentration range of 0.1 to 34 nM and detection limit of 0.04 nM using the SWV technique. The brilliant sensor performance was attributed to the high electrical conductivity of Ag@MNPs, fast electron transfer ability of G-nanosheets, and the large surface area of Fe_3_O_4_ NPs for the adsorption of antibodies in large amounts. The results approved the applicability of this electrode for kanamycin determination in animal food samples due to its good selectivity in presence of interferences (neomycin, gentamicin, vitamin C, and glucose), high stability, and very good sensitivity. Similarly, tetracycline was detected using a label-free electrochemical immunosensor fabricated via modification of Au NPs with magnetic NPs and using chitosan polymer as a linker [41]. Tetracycline (TET) was accurately detected over the concentration range of 0.1–2.26 nM and detection limit of 0.08 nM due to effective immobilization of monoclonal anti-tetracycline antibody over electrode surface and fast electron transfer kinetics. To ensure the sensor validity for detecting low-existed antibiotics in food samples, the sensor response was examined in presence of interfering structures like chloramphenicol, penicillin, gentamycin, and erythromycin. Additionally, the aptasensor principle was improved based on the use of antibodies (antibiotic biorecognition molecule) for stable, cost-effective, and highly precise electrochemical antibiotic sensing. In this context, the anti-tetracycline aptamer was immobilized effectively over a screen-printed electrode (SPE) modified with magnetic NPs and ionic liquid (IL) as shown in Figure 6a [89]. According to Figure 6a, the IL was used to modify clean and activated SPE surface to produce (IL/SPE) modified electrode that followed by casting Fe_3_O_4_ onto the modified electrode to obtain (Fe_3_O_4_/IL/SPE). Then, the aptamer was dropped onto the surface of Fe_3_O_4_/IL/SPE followed by the immobilization of TET to finally produce the aptasensor TET/Apt/Fe_3_O_4_/IL/SPE.

According to Fig6b, increasing TET concentration (from a to h that corresponding to the increase from 0.01 M to 1 × 10^−9^ M) led to the gradual increase of peak current. This linear proportion between peak current and TET concentration with a correlation coefficient of 0.9687 and linear slope of 0.888 indicated the ability to use this aptasensor in real sample applications. Tetracycline was detected effectively by the fabricated sensor over the concentration range of 1.0 to 108 nM with a detection limit of 1 nM using the voltammetric technique as shown in Figure 6b due to the fast and easy electron transfer between electrode/electrolyte interfaces that attributed to the interaction between IL and Fe_3_O_4_ NPs. During the analysis of tetracycline in milk samples, the sensor provided excellent selectivity, reproducibility, and stability. Further, a novel assay of magnetic bar carbon paste electrode was presented to improve the electrochemical sensing of tetracycline [90]. The magnetic nanocomposite of Fe_3_O_4_@oleic acid was deposited over a carbon paste electrode surface that was previously modified with magnetic bars. The best deposition of Fe_3_O_4_@oleic acid magnetic NPs produced an excellent response of the sensor. So, the sensor showed excellent response toward tetracycline over a detection range of 0.01 to 107 pM with a detection limit of <3.9 fM using impedimetric analysis that was highly improved compared to several previous studies. To encourage the use of a sensor for analysis of real samples (blood, drugs, honey, and milk samples), the sensor was tested for tetracycline detection in the presence of interferes include doxycycline and oxytetracycline with unaffected selectivity. Similarly, nanocomposites of nanoporous Pt–Ti alloy and Au NP–functionalized magnetic multi-walled carbon nanotubes provided a unique signal amplification concept for streptomycin precise electrochemical aptasensing [91]. In this study, PtTiAl alloy in hydrochloric acid was used for the construction of nanoporous Pt–Ti alloys via the Al selective de-alloying process. Using the DPV transduction method, the fabricated sensor over a detection range of 8 × 10^−5^ to 0.1 μM provided a detection limit of 0.01 nM. The electrode provided excellent recovery of streptomycin (97% to 105%) with reproducibility, stability, and selectivity in presence of interferes from real milk samples. Further, multiplexed electrochemical antibiotics sensing using magnetic NPs was potentially extended by the application of ExoI assisted cascade multiple amplification and metal ion encoded magnetic hollow porous nanoparticles [92]. In this study, the signal was magnified toward the detection of chloramphenicol and oxytetracycline antibiotics through the improvement of metal ions immobilization by using magnetic hollow porous nanoparticles as soluble carriers. Using the SWV technique, the resulted in low detection limit was 0.2 nM and 0.4 nM for oxytetracycline and chloramphenicol, respectively due to the dual signal amplification concept. Additionally, this sensor was found to be promising for oxytetracycline and chloramphenicol detection in milk samples especially, its electrochemical wave unaffected by the existence of interfering antibiotics include streptomycin, kanamycin, tetracycline, and doxycycline. Lately, streptomycin antibiotics highly specific recognized through magnetic molecularly imprinted polymers [93]. These authors fabricated nanospheres of magnetic molecularly imprinted polymers using gold metal ions that effectively enhanced molecularly imprinted polymerization on magnetic beads with a streptomycin template. Molecular imprints occur upon the adding of streptomycin due to the competitive-type strategy including the competition between glucose oxidase labeled streptomycin and the analyte and hence the detection process takes place. Thus, this signal amplification strategy resulted due to bioelectrocatalytic reaction to show a high sensitivity toward streptomycin over the concentration range of 0.09–34 nM with a detection limit of 0.01 nM. This sensor has a good recovery of 82% and 130% and selectivity during the electrochemical detection of streptomycin in the presence of tetracycline and chloramphenicol in real samples (honey and milk). After that, magnetite nanoparticle-modified molecular imprinted polymers were used to fabricate highly precise label-free nano-enabled devices for the detection of sulfonamides over the concentration range of 0.1 to 108 nM with a detection limit of 1 pM [94]. The presence of interfering antibiotics (sulfacetamide and sulfadiazine) was not affected the sensitivity of sulfonamide detection in real samples of spiked seawater. Similarly, methacrylic acid as a functional monomer was used to develop a kanamycin templated label-free MIP electrochemical sensor for the detection of kanamycin over the concentration range of 1 × 10^−4^ to 1.0 μM with a detection limit of 24 pM [95]. Additionally, the sensor has high selectivity toward familiar antibiotics such as erythromycin, gentamycin, and streptomycin due to the presence of kanamycin-specific imprinted binding sites that owing arrangement and size complementary to kanamycin functional groups. Direct electrocatalytic effect during antibiotic sensing can be conveniently induced via magnetic nanostructures. In this context, cefixime voltammetric detection was achieved via using multiwalled NiFe_2_O_4_ decorated carbon nanotubes as a mediator [96]. In this method, there was no bio-recognition element that makes this technique cost-effective and promising for application. Using linear sweep voltammetry as a transduction method, this sensor provided a linear range of 0.1 to 601 μM with a detection limit of 0.03 μM toward the detection of cefixime. The authors confirmed the potential applicability of the sensor toward cefixime detection in plasma, urine, and pharmaceutical samples especially after its unaffected sensitivity in presence of interferes (tartaric acid, CO_3_^−2^, SO_4_^−2^, NH_4_^+^, and Ca^+2^). Likewise, trace amounts of rifampicin in pharmaceutical and biological samples were detected using polyvinyl pyrrolidone capped CoFe_2_O_4_@Cd Se core-shell modified electrode [97]. Polyvinyl pyrrolidone capped CoFe_2_O_4_@Cd Se core-shell NPs have a high adsorptive capacity, abundant active sites, and large surface area that induced electrocatalytic activity of rifampicin. Using SWSV as a transduction method, the sensor provided high sensitivity toward rifampicin with a detection limit of 0.05 fM over the concentration range of 1 × 10^−4^ to 106 pM. During rifampicin analysis in pharmaceutical drugs and serum samples, the fabricated sensor provided brilliant selectivity in presence of familiar interferes like L-threonine, uric acid, pyrazinamide, and isoniazid. Lately, ciprofloxacin antibiotic was detected over carbon paste electrode modified using chitosan-coated Fe_3_O_4_ magnetic NPs (enable fast electron transfer accompanying the redox process of antibiotic) [98]. Using DPV as a signal transduction method, the sensor provided sensitivity toward ciprofloxacin over a detection range of 7 to 76 μM with a detection limit of 0.02 μM. They can be applied for urine and serum detection of ciprofloxacin with good recovery (96% to 106%), high sensitivity, and high selectivity.

#### 2.2.5. Carbon Nanomaterials

The properties of carbon nanomaterials include graphene, graphene quantum dots, fullerene, carbon nanofibers, and carbon nanotubes attracted the attention of scientists. Due to the different hybridization states of carbon-containing sp^3^, sp^2^, sp, and sp^2^-sp^3^, it is considered a very exciting element [99]. Carbon nanomaterials are considered to be promising for biosensor applications due to their magnificent properties such as robust mechanical strength, high electrical conductivity, biocompatibility, aspect ratio, and high chemical stability. The varied synthesis ways of carbon-based nanomaterials are responsible for their extraordinary electrochemical characters. Carbon nanomaterials have many morphological classes like zero to three-dimensional scales that are used in conventional biosensing as nanoprobes. So, the electrochemical biosensors based on carbon nanomaterials and applied for the detection of antibiotics are discussed in the next section.

##### Graphene-Based Nanomaterials

The sensor societies and electronics attract attention toward the application of 2D carbon nanomaterial (graphene) in their installation. Compared to single-walled carbon nanotubes (SWCNTs), single graphene nanosheets have better sensitivity, stability, higher surface area, and sixty folds more conductivity [100]. Furthermore, owing to its great potential window for high electron transfer speeds, graphene has a low charge-transfer resistance and increased electrochemical efficiency. But, Van der Waals forces and π-π bonds between graphene sheets cause the agglomeration of graphene nanomaterials that hinders their use in different applications. This limitation can be overcome through their modification to prevent their agglomeration and can be used in different applications. The application of graphene in the antibiotics electrochemical sensors attracted attention recently due to high specific surface area and good electron transfer [101]. Graphene is used to modify screen-printed carbon electrodes to enhance the electrocatalytic activity of the electrode for tetracycline detection [102]. Tetracycline was precisely detected using this sensor in milk and honey samples at very low levels of antibiotic (0.07 μM). Devices were developed further; chloramphenicol was sensitively detected using sensors of N-doped graphene nanosheet/Au nanoparticles composite [103] over the concentration range of 2.00 to 80.00 μM with a detection limit of 0.60 μM using LSV method. The Au/N-G electrochemical response towards chloramphenicol antibiotic was experienced in the existence of different interfering structures like K_2_S_2_O_8_, oxytetracycline, chlortetracycline, metronidazole, ascorbic acid, uric acid, glucose, and nitrobenzene. Therefore, chloramphenicol eye drops as a real sample approved the sensor applicability for chloramphenicol analysis in real samples. In another study, a water-soluble graphene sheet/Prussian blue chitosan/nanoporous gold biocompatible composite was used to detect kanamycin as a label-free electrochemical immunosensor [40]. Electrostatic adsorption was responsible for the modification of the electrode with Prussian blue (electron transfer mediator) and water-soluble graphene sheets. Kanamycin was sensed with a very low detection limit of 0.02 nM due to eased electron transfer resulted from brilliant conductivity of combined nanoporous Au and water-soluble graphene sheets. Besides excellent sensitivity, the sensor performance was unaffected by the presence of interfering antibiotics including neamine, neomycin, and gentamicin. After that, a novel signal amplification strategy was used for the fabrication of aptasensor for the electrochemical ultrasensitive detection of kanamycin antibiotic [104]. Thionine functionalized graphene and hierarchical nanoporous PtCu alloy for the first time were used as biosensing substrates. The high-affinity combinations between Pt NPs and amino groups over aptamers are responsible for the specificity of the aptamer toward kanamycin. Kanamycin was detected accurately by this aptasensor with a very low detection limit of 0.87 pM over the concentration range of 0.002–100 nM. Apart from exhibited high sensitivity, kanamycin was detected also with excellent selectivity in the presence of various interferences including tyrosine, dopamine, and human chorionic gonadotropin approved its applicability for the detection of kanamycin in real animal food samples. Ionic liquids/hematein/penicillinase adsorbed with graphene nanosheets films that assembled layer by layer for the detection of penicillin antibiotic [105]. The sensor detection theory was to use the penicillinase enzyme to transform penicillin into penicillin acid catalytically and to track the released H^+^ using hematein as a pH meter. Ultra-low penicillin detection was achieved using this sensor with a detection limit of 0.1 pM. The fabricated sensor demonstrated satisfied precision for detecting actual milk samples of penicillin without possible interfering by antibiotics such as levofloxacin, streptomycin, kanamycin, and ampicillin. Herein, the great IL conductivity and the capability of the film to form are responsible for the brilliant electrochemical response that maintains immobilized penicillinase favorable orientation over the electrode surface. Similarly, the high selectivity of molecularly imprinted polypyrrole was used to fabricate levofloxacin antibiotic sensor [106]. The electrocatalytic activity toward levofloxacin oxidation was enhanced through the incorporation of Graphene-gold composite with molecularly imprinted polypyrrole. The solubility of graphene was enhanced by the modification before graphene-Au synthesis with poly(diallyl dimethylammonium chloride) to ease the functionalization with gold NPs for the effective detection of levofloxacin. Using a DPV transduction method, the sensor showed a detection limit toward levofloxacin of 0.50 μM over the concentration range of 1.01 to 100 μM. During levofloxacin detection in pharmaceutical capsules, the sensor showed excellent reproducibility, selectivity, and sensitivity. Later, reduced graphene oxide 3-D architectures were used to fabricate template-free and green strategies for the detection of chloramphenicol [107]. This sensor showed enhanced electrochemical response toward chloramphenicol with a detection limit of 0.16 μM compared to previously reported methods. Moreover, the electrical conductivity and active sites were increased through the synthesis of Au/carbon nitride/graphene nanostructured 3-D composite for the potential detection of ciprofloxacin and chloramphenicol antibiotics [43]. In the presence of ciprofloxacin, the authors supposed an excellent electrochemical response to chloramphenicol with a detection limit of 28 nM. Moreover, in milk samples with adequate recoveries ranging from 96.0 to 101.0 percent, the fabricated sensor was also found to be applicable for the experimental quantification of chloramphenicol. Moreover, in recent years, the inherent exceptional physical and chemical characteristics of metal oxides have made them hopeful for various biosensors applications. For example, chloramphenicol was detected using an electrochemical sensor fabricated hydrothermally from reduced graphene oxide and nanocrystals of cobalt oxide [108]. Chloramphenicol was detected over the fabricated sensor accurately with a sensitivity of 1.34 μA μM cm^−2^ and a detection limit of 0.60 μM. The sensor’s potential was further improved, 3D hierarchical zinc oxide was used for the modification of graphene oxide to fabricate chloramphenicol electrochemical sensor [109]. graphene oxide/ZnO nanocomposite sensor provided high sensitivity detection of chloramphenicol of 7.30 μA μM^−1^cm^−2^ with a detection limit of 0.02 μM over the concentration range of 0.20 μM to 7.30 μM that found improved compared to previous studies. The sensor potential against commonly present structures such as glucose, uric acid, and ascorbic acid was tested and confirmed the applicability of graphene oxide/ZnO nanocomposite sensor in real sample applications. In a later work, there was an attempt to improve graphene electrochemical behavior through doping with heteroatom [110]. Subsequently, reduced graphene oxide was doped using chlorine to fabricate a novel electrochemical sensor for the detection of pharmaceutical samples, calf plasma, milk, water, and veterinary drugs containing chloramphenicol. The study reported that the facilitated sensitive determination of chloramphenicol in veterinary drugs attributed to the enhanced electrochemical conductivity and electrocatalytic activity of chlorine doped reduced graphene oxide. Using the DPV transduction technique, the sensor showed excellent sensitivity toward chloramphenicol over the concentration range of 3 to 35 μM with a detection limit of 2 μM. Additionally, the efficiency of the sensor was unaffected with interfering antibiotics like tetracycline, erythromycin, and penicillin G. Levofloxacin voltammetric detection was achieved using a new concept containing composites of poly (o-aminophenol) and graphene quantum dots [57]. The sensor showed high sensitivity toward levofloxacin determination over the concentration range of 0.06 to 100 μM with a detection limit of 11 nM due to the excellent electrocatalytic activity of the composite. Additionally, the sensor found proper for levofloxacin detection in milk due to its brilliant recovery that reached 95% to 101%. Trace levels of streptomycin were detected using a talented biomimetic strategy using composites of streptomycin imprinted poly(pyrrole-3-carboxy acid) and electrochemically reduced graphene oxide [111]. The fabricated composite showed excellent molecular recognition towards streptomycin in porcine kidney and honey samples with reliable scalability and a very low detection limit of 0.6 nM. Additionally, gatifloxacin was electrochemically determined using composites of reduced–graphene oxide and β-cyclodextrin [112]. To get the maximum response toward gatifloxacin antibiotic, the authors studied the effect of several factors like accumulation potential, electro-polymerization cycles, and pH to finally reach a sensitivity of 0.34 μA μM^−1^cm^−2^, and detection limit of 0.01 μM over a concentration range of 0.06 to 151 μM toward the antibiotic. The sensor was found very applicable for gatifloxacin detection in human urine due to its improved long-term stability, reproducibility, and selectivity. In another report, midecamycin was accurately determined using a sensor fabricated from reduced graphene [113]. Compared to GCE, the electrochemical response of the modified electrode toward midecamycin was improved. The sensor revealed good sensitivity toward midecamycin with a detection limit of 0.2 μM over the concentration range of 0.4 to 200 μM. The sensor was found applicable for midecamycin detection in real samples of pharmaceutical and biological fluids due to its acceptable recoveries. As an extension of that study, trace amounts of sulfamethoxazole were detected via a novel concept comprising the association of graphene oxide, ionic liquids, and metal oxides [114]. According to the results, sulfamethoxazole was detected effectively in tablet and urine real samples due to the nanocomposites’ cumulative effect and improved electrocatalytic activity.

##### Carbon Nanotubes

Carbon nanotubes (CNTs) were firstly introduced in 1991 by Iijima as carbon atom sp^2^ hybridized rolled graphene sheets to form 1-D carbon-based nanomaterials [115]. Their electronic, thermal, and mechanical characteristics such as unique thermal conductivity, excellent electrochemical stability, fast electron transfer, and great mechanical flexibility attracted attention for their application in different fields especially in the electrochemical biosensors [116]. However, CNTs’ low solubility, on the other hand, stymies their development. As a result, the process of fabrication of CNT-based composites is crucial. This limitation of CNTs can overcome by their functionalization. Various functional groups can easily functionalize CNTs that allow their linkage to different organic molecules and biomolecules and make them very proper to recognize biomarkers, metabolites, and analytes. In the last years, CNTs-based biosensors led to an improvement in electrochemical potential with a low detection limit and broad detection range for the detection of antibiotics. Levofloxacin in human urine was electrochemically detected using a sensor fabricated from vertically aligned CNTs [117]. Levofloxacin in urine was determined using this fabricated sensor over the concentration range of 1 to 11 μM and the limit of detection of 75 nM indicated the brilliant electrocatalytic activity of vertically aligned CNTs against electrooxidation of levofloxacin. Similarly, an electrochemical aptasensor based on signal amplification strategy used for tetracycline detection on multi-walled carbon nanotubes functionalized with carboxyl groups [118]. The excellent biocompatibility and conductibility of multi-walled carbon nanotubes led to the detection of tetracycline in spiked milk samples with a low detection limit of 6 nM. During the analysis of real milk samples, the sensor electrochemical response was examined in the existence of tetracycline derivatives including doxycycline hydrochloride and oxytetracycline hydrochloride. The performance of sensors was further improved toward tetracycline detection via the synthesis of aptasensor by the combination of multiwalled carbon nanotubes-chitosan and chitosan-Prussian blue-graphene nanoparticles [119]. Tetracycline in spiked milk samples was detected sensitively with a detection limit of 6 pM over the concentration range of 1 × 10^−4^ to 104 μM due to graphene and multiwalled carbon nanotubes excellent synergy. The sensor was tested against interfering molecules like gentamycin, kanamycin, and oxytetracycline before its use for detection of tetracycline in cattle products. Metal NPs’ brilliant electronic properties have been commonly utilized to fabricate electrochemical sensing concepts against several antibiotics in combination with multi-walled carbon nanotubes. In this context, adriamycin was detected using an electrochemical sensor fabricated from nanocomposites of Ag NPs and carboxyl functionalized multi-walled carbon nanotubes [120]. Using DPV as a transduction technique, adriamycin was electrochemically detected using the modified electrode with the enhanced response over the concentration range of 8.0–20 nM and detection limit of 2 nM. During the identification of calf thymus DNA, the sensor was shown to have excellent specificity, reproducibility, and stability. In a study, urine and blood valrubicin was detected using an electrochemical sensor fabricated from multiwalled carbon nanotubes and Au NPs [121]. This study on the analytical determination of valrubicin after its discovery in the year 1998 was reported as a “starting point” toward its detection. The effective immobilization of several sequences of ss-DNA probes on the surface of the composite was responsible for the different interactions between valrubicin and DNA as reported in this study. Thus, the attraction to valrubicin was enhanced by using guanine and cytosine sequences. The fabricated sensor showed valrubicin high sensitivity over a concentration range of 0.6 to 81.0 μM with a limit of detection equal to 0.019 μM. Further, the sensor electrochemical response toward valrubicin in human serum and urine was tested in the presence of common interferences including K^+^, Na^+^, Mg^+2^, glucose, urea, and sucrose. A tetracycline sensor was fabricated using molecularly imprinted polymer incorporated with Au NPs modified multiwalled carbon nanotubes [122]. Improved ability of electron transfer to the electrode surface from imprinted sites that were found with sufficient amount over the composite electrode was reported here. Au NPs’ excellent electrocatalytic activity is mainly responsible for the tetracycline sensing with a very low detection limit of 89 nM as resulted using the CV transduction technique. Sensor selectivity was tested in the existence of familiar interferes like chloramphenicol, oxytetracycline, and nafcillin while the authors of this analysis reported that oxytetracycline caused more interference during tetracycline detection than either chloramphenicol or nafcillin. After that, neomycin was detected over an imprinted electrochemical sensor based on the Au electrode decorated with chitosan-silver nanoparticles/graphene-multiwalled carbon nanotubes composites [123]. Electropolymerization was used neomycin as a template for the preparation of the molecularly imprinted polymers while the monomer used is pyrrole. To achieve the maximal activity of the imprinted sensor towards neomycin, the effect of different factors was studied like incubation time, elution time, and template/monomer ratio. The combined properties of chitosan-silver nanoparticles and graphene-multiwalled carbon nanotubes caused the high sensitivity and high binding affinity of the sensor toward neomycin and get detection limit of 7.60 nM. The sensor was found a promising cost-effective method for neomycin detection in milk and honey real samples due to its unaffected electrochemical response in the existence of polycyclic compounds and neomycin analogs such as kanamycin sulfate, streptomycin, and gentamycin sulfate. Cefotaxime was detected using a sensitive voltammetric system fabricated using the electrodeposition of bimetallic Au–Pt NPs over multiwalled carbon nanotubes [124]. The sensor showed high sensitivity toward cefotaxime over the concentration range of 4 × 10^−3^ to 10 μM with a limit of detection equal to 1.00 nM that was higher than that of GCE modified separately either by Au–Pt bimetallic NPs or multi-walled carbon nanotubes. The cefotaxime effective accumulation at the surface of the electrode and multi-walled carbon nanotubes/Au-PtNPs effective electrochemical surface area is responsible for the enhanced electrochemical response of the sensor. This fabricated sensor showed clear electrochemical advantages such as anti-fouling behavior, repeatability, and reproducibility besides revealed three orders of magnitude. Accordingly, pharmaceutical cefotaxime can be detected using this validated sensor. Chloramphenicol was detected using the nanohybrid materials-based sensor in which molecularly imprinted polymer was coated via layer by layer assembly on the surface of Cu NPs deposited on carbon nanotubes [125]. The detection limit showed by the sensor toward chloramphenicol was superior and found to be 11 μM. Additionally, the chloramphenicol selectivity of this fabricated sensor was 100% even in the existence of interferences molecules like thiamphenicol, clindamycin, dansyl chloride, and florfenicol. The specificity of the molecularly imprinted polymer network, Cu NPs catalytic activity, and the excellent carbon nanotubes conductivity are responsible for the improved sensitivity of the fabricated sensor. Sulfamethoxazole and trimethoprim antibiotics were simultaneously detected using mercury-free electrochemical sensing concept through antimony NPs modified multi-walled carbon nanotubes that formed a paraffin composite electrode [126]. During this detection of sulfamethoxazole and trimethoprim in environmental samples, it was found that antimony NPs addition onto multi-walled carbon nanotubes had a synergistic impact on the response toward target antibiotics. The fabricated sensor exhibited detection limits of 30 nM and 25 nM toward trimethoprim and sulfamethoxazole, respectively. The sensor efficiency was examined toward the two antibiotics in organic wastewater. Thus, under optimized conditions, the fabricated sensor during the analysis of natural water showed good recoveries of 97% to 102%. Further, An electrochemical sensor associated with multiwalled carbon nanotubes, conducting polymer poly(diphenylamine), and cetyltrimethylammonium bromide (CTAB) surfactant was fabricated to sense chloramphenicol in biological samples and benefit the properties of multiwalled carbon nanotubes and their metal NPs composites [127]. Using cyclic voltammetry, the results revealed that the reduction of chloramphenicol over the electrode modified was adsorption controlled [128]. To achieve a wide detection range, accepted selectivity, and good stability toward chloramphenicol sensing, the authors tested the impacts of different factors like surfactant and nanotubes amount, electrolyte solution, scan rate, and accumulation conditions.

##### Other Carbon Nanomaterials

To advance the potential and detection limit of the electrochemical sensor toward antibiotics, many efforts have been paid for the synthesis of novel carbonaceous nanomaterials due to their excellent adsorption ability and biocompatibility. In this context, the mesoporous carbon-silicon nanocomposites co-doped with iron-nitrogen was used for the modification of glassy carbon electrode to provide chloramphenicol label-free sensor [129]. The sensor provided an enhanced response toward chloramphenicol with a detection limit of 0.04 μM over the concentration range of 1.00 to 500 μM due to the excellent electrocatalytic characters of the nanocomposite. Additionally, the sensor’s excellent reproducibility, stability, and selectivity allowed its applicability for chloramphenicol determination in eye drop samples. Chlortetracycline residues were determined using a semi-derivative, reliable, and simple concept that advanced using signal amplification of yolk-shell structured carbon sphere@MnO_2_ [130]. A facile redox reaction between carbon spheres and MnO_4_ in an acidic medium was used to fabricate a novel yolk-shell structure. The low density of the composite and excellent accumulation ability allowed the sensor to detect chlortetracycline with a detection limit of 0.27 μM over the concentration range of 0.4 to 300 μM using the DPV method. This sensor was successful in food samples chlortetracycline determination with excellent tolerances against five-fold concentrations of tetracycline, rifampicin, and oxytetracycline, 1500-fold concentrations of glycine, glucose, and Al^+3^, and 3000-fold concentrations of K^+^ and Na^+^. Moreover, tetracycline was detected based on a ratiometric electrochemical aptasensor simple strategy [131]. To immobilize anti-tetracycline aptamers, two separated aptamers were used one is fabricated based on ferrocene and gold NPs, and the other aptamers based on carbon nanofibers and gold NPs. The sensor was a good choice to solve the issue of large batch variations and low precision. This sensor showed a low detection limit toward tetracycline of 0.5 nM over the concentration range of 2 × 10^−5^ to 2 μM. In another study, carbon black-dihexadecyl phosphate as a cost-effective composite was used for the simultaneous detection of amoxicillin antibiotic in the existence of inflammatory drug nimesulide [132]. Using the SWV technique, the sensor exhibited an improved response toward amoxicillin with a detection limit of 0.13 μM. A novel label-free electrochemical aptasensor was constructed using TiO_2_ and SnOx nanocrystals embedded in mesoporous carbon nanospheres for the detection of antibiotic tobramycin [133]. A detection limit of 0.02 nM was achieved using that sensor due to the sensor’s high affinity towards tobramycin-targeted aptamers, accepted electrochemical activity, and large surface area. The sensor was found applicable for the detection of human urine and serum tobramycin over the wide concentration range of 0.03 to 11 nM using electrochemical impedance spectroscopy (EIS) as an analysis technique with enhanced selectivity and sensitivity. Similarly, oxytetracycline was determined using a label-free electrochemical sensor fabricated from the calcination of Fe (II) based metal-organic frameworks to synthesize Fe_3_O_4_@mesoporous carbon nanocomposite [134]. An efficient immobilization skeleton was saved to strands aptamers of oxytetracycline due to high stability, chemical functionality, and strong bio-affinity of Fe_3_O_4_@mesoporous carbon nanocomposite that finally provided a wide detection range of 0.02 to 2.00 nM and a very low detection limit of 5 × 10^−5^ nM toward oxytetracycline antibiotic by using impedimetric studies. The sensor demonstrated brilliant anti-interference potential in milk samples to undergo oxytetracycline identification in the existence of chlortetracycline, doxycycline, and tetracycline. Moreover, pyrazinamide was determined using a label-free electrochemical sensor fabricated using nanodiamond as a sensitive carbonaceous nanomaterial [135]. The sensor is cost-effective as it is not required costly signal labels or tags while provided a very low detection limit of 221 nM over the concentration range of 0.8 to 50 μM for pyrazinamide detection. Lately, kanamycin and streptomycin were electrochemically detected using multiplexed aptasensor fabricated from composites of mesoporous carbon-gold nanoparticles and carbon fibers [35]. The presence of lead sulfide and cadmium sulfide as amplification tags enhanced the sensitivity toward kanamycin and streptomycin and provided detection limits of 87 and 46 pM, respectively. The multiplexed sensor exhibited a concentration range of 0.2 to 10^3^ nM toward both antibiotics using the DPV detection method. The excellent electron transfer kinetics and high electronic conductivity resulted from the synergistic effect of the combined mesoporous carbon-gold nanoparticles and carbon fibers are responsible for the improved electrochemical response of the aptasensor. In another study, streptomycin was electrochemically detected using multifunctional graphene nanocomposites and porous carbon nanorods [136]. An improved detection limit of 0.05 nM was achieved toward streptomycin due to the existence of amplification tags represented by multifunctional graphene nanocomposites. The sensor was found applicable for the determination of streptomycin in real milk samples due to its good selectivity, great sensitivity, high reproducibility, and wide concentration range of 0.09 to 341 nM via using DPV as a detection technique.

## 3. Evaluation of the Potential of Different Nanomaterials-Based Sensors

Due to their ease of use, cost-effectiveness, excellent selectivity, and inherent flexibility, electrochemical sensors as a powerful analytical tool have been recognized [137]. The efficiency of such sensors, however, depends significantly on the type of antibiotics, the method of transduction, and the choice of materials used in the sensing procedure. Thus, the antibiotic detection efficiency using nanomaterial-based electrochemical sensors is contrasted with various quality assurance factors (like selectivity, sensitivity, detection range, and limit of detection) toward different antibiotic pollutants. Multi-walled carbon nanotubes modified sensors (anti-TET/MWCNTs/GCE) thus, revealed excellent sensitivity toward tetracycline antibiotics with a detection limit of 6 nM as shown in Table 2. Using a modified electrode with nanocomposites of multi-walled carbon nanotubes and chitosan-Prussian blue-graphene (Anti-TET/Gr/(CS-PB-GR)2/MWCNTs-CS/GCE), the efficiency of this device was further enhanced to 0.006 nM. Methylene blue modified Au NPs (MB/Au NPs) displayed a comparable detection limit of 5 pM toward antibiotic sensing, which was greatly enhanced by the utilization of magnetic nanoparticles (Fe_3_O_4_ nanoparticles@OA) to achieve a tetracycline detection limit of 4 fM. The existence of well-defined large specific surface area and pore sizes, however, permitted MMOF-MIP to display an enhanced detection limit (0.23 fM) against tetracycline sensing. According to that, relative to the current nanomaterials used for tetracycline detection, MMOF-MIP is regarded as a leader applicant for electrochemical tetracycline sensing. Pt nanoparticles, although, demonstrated a moderately great limit of detection (4281 nM) against tetracycline antibiotics. So, the data indicated that nanoparticles of noble metals must be additionally changed in the future to serve as a sensitive tetracycline detection material. Carbon-based NMs have acted as an economical alternative to electrochemical chloramphenicol detection with extraordinary chemical and physical characteristics as shown in Table 2. Multi-walled carbon nanotubes composites (MIL-101(Cr)/XC-7) showed a very low detection limit of 1.6 nM against the sensing of chloramphenicol, which was a competitor to that achieved using MWCNT-CTAB-PDPA (detection limit of 2 nM). Alike, kanamycin was detected using electrochemical sensors based on different NMs such as MOFs, magnetic NMs, metal NPs, and carbonaceous NMs as shown in Table 2. According to that, kanamycin was detected with a very low detection limit of 0.014 nM reached using the composite of nanoporous Au, Prussian blue-chitosan, and water-soluble graphene sheet. By the presentation of hierarchical nanoporous Pt-Cu alloy, and thionine functionalized graphene as a glassy carbon electrode modifier, this detection limit was advanced to 0.90 pM. Nanomaterials, although revealed a higher limit of detection toward the sensing of kanamycin antibiotic than that of equivalent nano-metal organic frameworks/biocodes that exhibited a detection limit of 0.00017 nM. So, the large surface area and simple functionalization procedures (using antibodies, aptamers, and metal NPs) made the functionalized graphene a promising kanamycin electrochemical sensing applicant. But, according to different sensing systems, nano-metal organic frameworks/biocodes sensors with a detection limit of 0.00017 nM against kanamycin appeared to be highly sensitive electrodes compared to the graphene-based sensors. Lastly, streptomycin electrochemical sensors based on the discussed nanomaterials were also, summarized in Table 2. Streptomycin was detected in different media (H_2_O, urine, milk, eye drop, and honey) with a wide detection range, low detection limits, good stability, high sensitivity, and excellent reproducibility due to the use of nanomaterials. Until now, streptomycin detection reports by any transducer materials have not reached the detection limit value achieved by NP-PtTi/Au@MWCNTs–Fe_3_O_4_ (0.02 nM). Take a comprehensive look at nanomaterials-based sensors’ potential toward the detection of antibiotics reveals that nanomaterials and their composites play a significant role in the decreasing of detection limit toward sensing of antibiotics and accordingly, enhancing the system sensitivity. Kanamycin and tetracycline were detected with a very low detection limit achieved using NMOF/biocodes and MOFs, respectively due to the porous nature of these materials. Likewise, chloramphenicol was also detected sensitively using carbonaceous porous nanomaterials. From the previous results, it is indicated that the lowest detection limit of a sensor toward antibiotic electrochemical sensing was that fabricated based on MOFs with a detection limit of 0.24 fM against tetracycline sensing followed by that based on magnetic nanomaterials. But, magnetic nanomaterials-based sensors require special experimental conditions that hinder their use in different applications. These low detection limits provided by MOFs-based sensors may be attributed to the high porosity of MOFs that save large surface area (10,000 m^2^.g^−1^) for modification to proper the characters of the target analyte. MOFs have the best biocompatibility than other nanomaterials that made MOFs the best for in vitro and in vivo analysis of analytes. But, all other nanomaterials-based sensors still promising for antibiotics detections with very low detection limits compared to the classical electrochemical sensors. Each nanomaterial-based sensor is not effective for all analytes but selected depending on several factors like the structure of the target antibiotic. Consequently, antibiotics ultra-sensitive sensing depends on many factors include antibiotics structure, large surface area, and the porous nature that are the vital considerations for the fabrication of electrode transducers. Additionally, multicomponent composites can be produced to reach an ultra-low detection toward residues of antibiotics by pay efforts to create a simple functionalization technique. It is supposed to determine the thermodynamic parameters (as ΔH, ΔS, and ΔG) [138,139,140,141,142,143] of the electrochemical reaction to getting a full image of the used sensor in the detection of antibiotics.

## 4. Conclusions

Antibiotics possess a serious threat to human safety and health through the accumulation in the human body across the food chain. As a result, developing responsive and convenient methods for determining antibiotic levels quickly is highly desirable. This review provides a summary of the latest improvements and applications of nanomaterial in the field of biosensors toward the determination of antibiotics in different systems with a discussion of their role in the improvement of sensor performance as indicated from linear detection range, selectivity, and sensitivity. It has been shown that electrochemical sensing techniques developed with the help of nanomaterial technology have improved the efficiency of the sensor with sensitive, precise, and quick antibiotic detection. A deep comparison look of different nanomaterials-based sensors has shown that sensor sensitivity can be dramatically increased across different routes such as the modification of the surface design, construction of hybrid materials, and chemical functionalization. However, additional advances in this field of research need to be done to overcome the significant challenges in this field of study that contain similar antibiotics cross-interference (as chlortetracycline, tetracycline, and oxytetracycline), biorecognition molecules, and nanomaterials biocompatibility, find the relationships between nanomaterials reactivity and the structure, fabrication of new functional materials, and controlled fabrication (surface area, shape, and size). Consequently, it is anticipated that the consumption of nanomaterials-based electrochemical sensors can provide an extremely dependable on-site study of antibiotics in real-world conditions and diverse biological ecosystems. By achieving synergies between electrochemical methods, recognition elements, and multifunctional nanomaterials, the technical difficulties accompanying these improved sensing devices can be further resolved. The desired movable sensors can be built through successful networking and electrochemical sensor combination via communication tools like tablets and smartphones. Successful convergence of two separate study fields is believed to have tremendous potential to successfully revolutionize all accessible antibiotic analytic routes.

## Figures and Tables

**Figure 1 micromachines-12-00308-f001:**
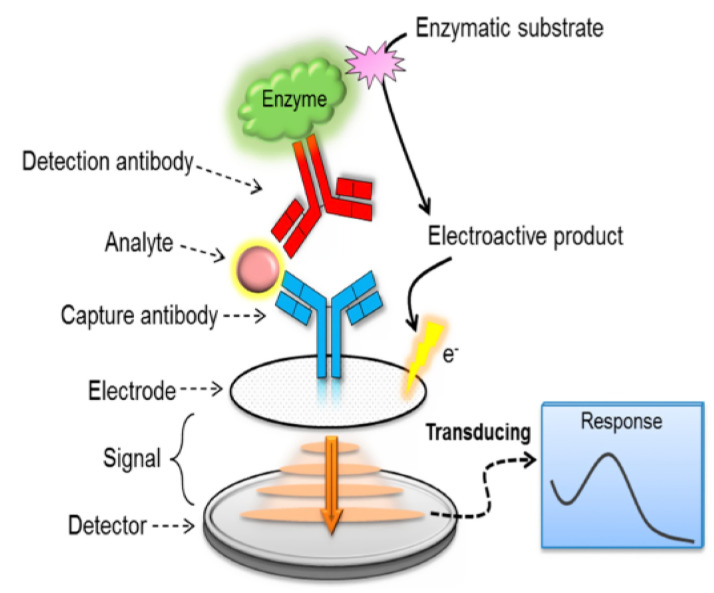
Electrochemical immunosensor basic analytical principle. Reproduced from MDPI [47].

**Figure 2 micromachines-12-00308-f002:**
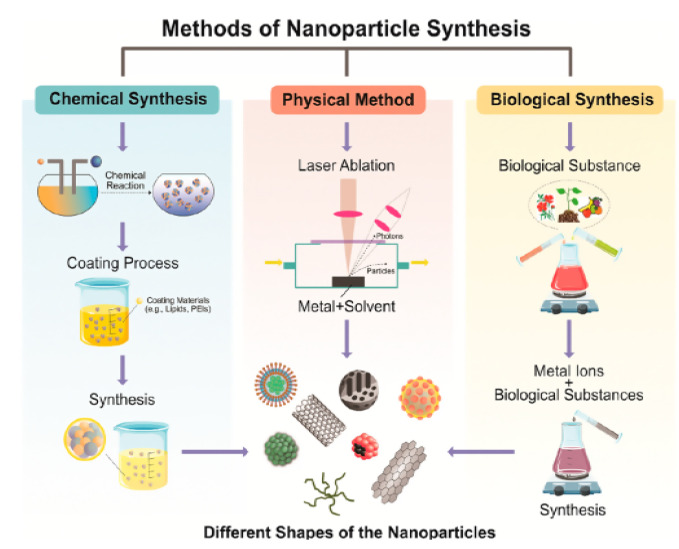
Classical methods of nanoparticle synthesis. Reproduced from MDPI [51].

**Figure 3 micromachines-12-00308-f003:**
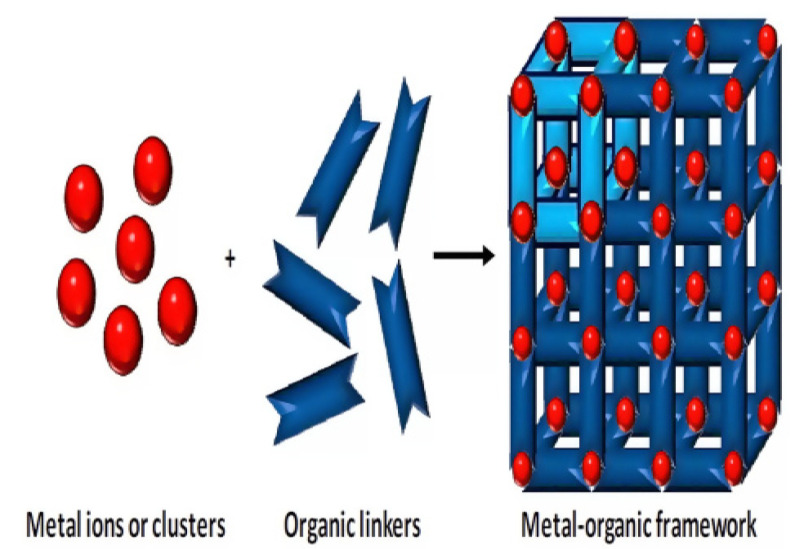
Metal-organic frameworks (MOFs) synthesis by mixing clusters and metal ions with organic linkers for the production of a cross-linked network. Reproduced from MDPI [60].

**Figure 4 micromachines-12-00308-f004:**
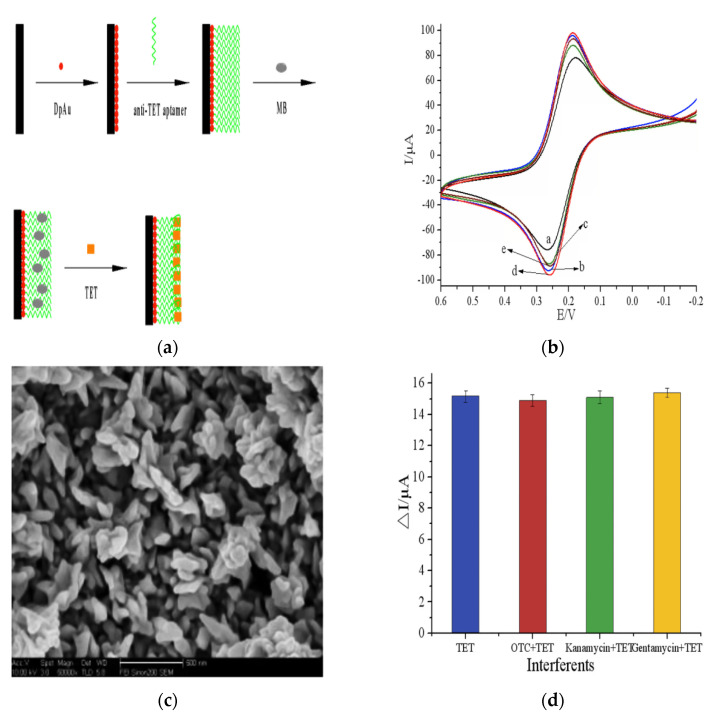
Methylene blue (MB)/anti-tetracycline (TET) aptamer/DpAu/glassy carbon electrode (GCE) immunosensor fabrication process (**a**), cyclic voltammograms (CVs) of different modification processes of GCE (**b**), scanning electron microscope (SEM) image of dispersed AuNPs over GCE (**c**), aptasensor specificity in the presence of different interferences (**d**). Reproduced from ESG [71].

**Figure 5 micromachines-12-00308-f005:**
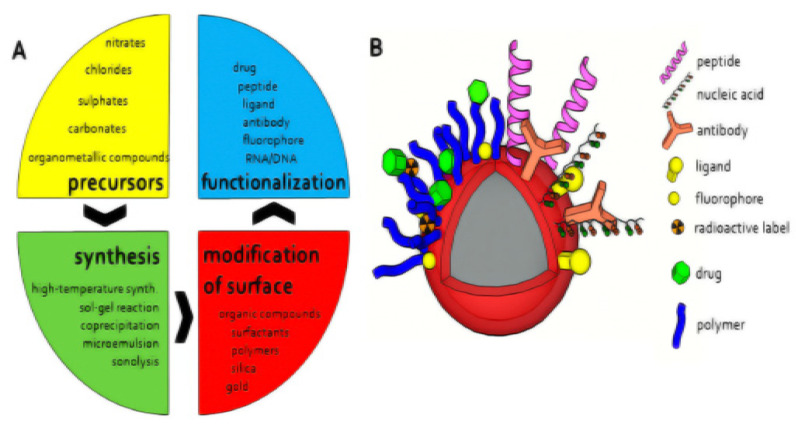
Design workflow (**A**), and possible functionalization (**B**) of magnetic nanomaterials. Reproduced from MDPI [85].

**Figure 6 micromachines-12-00308-f006:**
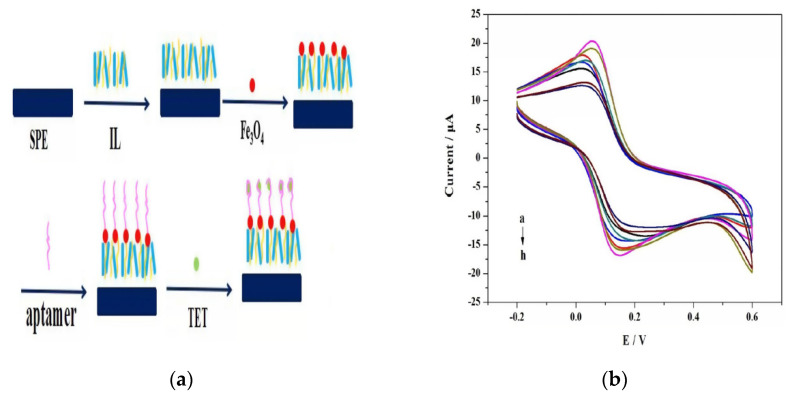
TET/Apt/Fe_3_O_4_/ionic liquid (IL)/screen-printed electrode (SPE) Aptasensor fabrication process (**a**), cyclic voltammograms of the aptasensor toward different concentrations of tetracycline (1.0 to 108 nM) (**b**). Reproduced from ESG [89].

**Table 1 micromachines-12-00308-t001:** The different electrochemical techniques used for sensing antibiotics.

Method	Principle	Limit of Detection	Applications
Electrochemical impedance spectroscopy	Small-amplitude sinusoidal AC excitation signal is applied to measure the resistive properties	10^−12^ M	Study of antigen-antibodies reaction, corrosion, and electron transfer kinetics
Chronoamperometry	The stepped potential is applied and the current measured	10^−5^ M	Measure electrode process mechanism, working electrode surface area, and analytes diffusion coefficient
Stripping technique	Worked electrode carries the pre-concentrated analyte then analyte stripped by application of scan potential from the electrode	10^−9^ M	Detection of trace elements
Square wave voltammetry	Current is determined as a consequence of square wave potential superposed on staircase waveform	10^−8^ M	Detection of trace elements, the study of catalytic homogeneous chemical reactions, and electrode kinetics
Differential Pulse voltammetry	Current is determined as a function of applied voltage superposed as regular voltage pulses superposed on the potential linear sweep or stair steps	10^−7^ M	Detection of trace elements
Linear Sweep Voltammetry	Voltage is applied then the current measured on the working electrode surface	10^−5^ M	Determination of analytes concentrations, unknown reactions, and irreversible reactions
Cyclic Voltammetry	Voltage is applied then the current measured on the working electrode surface	10^−5^ M	Assessment of reaction products, trace reaction intermediates, and study redox reactions

**Table 2 micromachines-12-00308-t002:** Antibiotics electrochemical sensors based on nanomaterials and their composites.

Electrode	Interface	Transduction Method	Antibiotics Detected	Limit of Detection (nM)	Selectivity	Real Samples	Ref.
Au	QD–cDNA2/cDNA1/ Cap-DNA	SWASV	Tetracycline, chloramphenicol, and streptomycin	20, 5, and 10, respectively	-	Milk	[23]
GCE	Dendritic probe encoded with magnetic aptamer QDs	SWV	chloramphenicol	0.001	Oxytetracycline and kanamycin	Fish	[37]
GCE	PoAP/GQD	LSV	Levofloxacin	10	Norfloxacin, lomefloxacin, enrofloxacin, and ciprofloxacin.	Milk	[57]
GCE	CdS QDs	DPASV	Ciprofloxacin	23	Gentamycin, erythromycin, kanamycin, chloramphenicol, and ofloxacin,	Human urine	[56]
Au	Ce-MOF@COF	EIS	Oxytetracycline	0.000036	Kanamycin, streptomycin sulfate, doxycyclinehyclate, bleomycin, and ampicillin	Urine, water, and milk	[66]
GCE	Y-DNA-NMOF	SWV	Oxytetracycline, and chloramphenicol	0.000049, and 0.000034, respectively	gentamicin sulfate, tetracycline, doxycycline, and kanamycin	Milk	[64]
GCE	NMOF Probe labeled with magnetic aptamer	SWV	Kanamycin and oxytetracycline	0.00016, and 0.00019, respectively	Gentamicin sulfate, doxycycline, streptomycin, chloramphenicol, and Chlortetracycline	Milk	[36]
GCE	Aptamer-metal ions NMOF Biocodes	SWV	Chloramphenicol and kanamycin	0.00020, and 0.00017, respectively	metal ions (K, Ca, Mg), oxytetracycline, and chlortetracycline	Milk	[63]
GCE	MIL-101(Cr)/XC-72	DPV	Chloramphenicol	1.6	Amikacin, gentamicin, neomycin, rutin, quercetin, penicillin, kanamycin, kitasamycin, tetracycline, and chlortetracycline	Milk, eye drop, and honey	[62]
Au	MMOF-MIP	LSV	Tetracycline	0.00000023	Doxycycline	Honey	[61]
GCE	CoFe_2_O_4_@CdSe capped with PVP	SWV	Rifampicin	0.00000005	Glucose, L-threonine, uric acid, pyrazinamide, and isoniazid	Pharmaceutical drug and serum	[79]
GCE	NiFe_2_O_4_-MWCNTs	CV	Cefixime	19	Ascorbic acid, glucose, tartaric acid, CO_3_^−2^, SO_4_^−2^, NH_4_^+^, and Ca^+2^	Plasma, urine, and tablets	[78]
CE	MMIP/CE	DPV and CV	Kanamycin	0.03	Erythromycin, streptomycin, and gentamycin	Milk and animal food derivatives	[77]
SPCE	MIP decorated Fe_3_O_4_ MNPs	EIS	Sulfamethoxazole	0.002	Sulfacetamide and sulfadiazine	Seawater	[76]
GCE	Aptamer/NP-PtTi/ Au@MWCNTs–Fe3O4	DPV	Streptomycin	0.02	Streptomycin, neomycin sulfate, kanamycin sulfate, and terramycin	Milk	[73]
MBCPE	Fe_3_O_4_ NPs@OA/antiTET	EIS	Tetracycline	0.000004	Doxycycline and oxytetracycline	Serum, honey, milk, and drugs	[72]
SPE	Fe_3_O_4_/IL	CV	Tetracycline	1.00	-	Milk	[71]
Au	Ab-MNPs-chitosan	DPV	Tetracycline	0.08	Chloramphenicol, penicillin, gentamycin, and erythromycin	Milk	[41]
GCE	TH-GS/GA/ Ag@Fe_3_O_4_-Ab	SWV	Kanamycin	0.04	Neomycin, gentamicin, vitamin C, and glucose	Animal foods	[70]
GCE	Pt NPs/C	DPV	Tetracycline	4281	-	Human urine	[97]
GCE	Pt Nps/MWCNT	LSV	Ceftriaxone	9.02	Lidocaine	Human serum	[98]
GCE	MIP/Ag@Au Nps/Ils	DPV	Ceftizoxime	0.003	Dopamine and ascorbic acid	Pharmaceuticals	[95]
GCE	Fe_3_O_4_-CMC@Au	SWV	Chloramphenicol	67.00	Ca^+2^, glucose, xanthine, cysteine, uric acid, and ascorbic acid	Urine	[96]
CPE	GNPs/MWCPE	SWV	Cefixime	4.00	Caffeine, glucose, oxalic acid, uric acid, citric acid, and ascorbic acid,	Tablets and human urine	[93]
ME	gold colloids	CV	Tetracycline	200	-	-	[94]
SPCE	Tyr-AuNPs	Amperometric	Sulfamethoxazole	22 × 10^3^	-	Water	[91]
Graphite SPE	HEM/Apt/AuNPs/ SBA-15@ DABCO	DPV	Chloramphenicol	5.00	Florfenicol, amoxicillin, cephalexin, and cefixime	Blood serum	[89]
Interdigitated Au SPE	Aptamer/AuNCs-Cys	SWV	Chloramphenicol	5.00	Florfenicol, amoxicillin, cephalexin, cefixime, and chloramphenicol	Human serum	[89]
GCE	MB/Anti-TET/AuNps	CV	Tetracycline	0.005	Gentamycin sulfate, kanamycin monosulfate, and oxytetracycline hydrochloride	Milk	[87]
MIP/NPAMR	MIP/NPAMR	CV	Metronidazole	0.00003	Dimetridazole, 4 -nitroimidazole, and 1,2 dimethylimidazole	Tablets and fish tissues	[86]
SPE	Aptamer/poly-DPB/ AuNPs	LSV	Kanamycin	9.5	Sulfadimethoxine, tetracycline, ampicillin, streptomycin, and neomycin	Milk	[83]
GCE	AuNPs/poly TTBA/ PS/aptamer/AuNPs	DPV	Daunomycin	0.053	Adriamycin, anthraquinone, neomycin, chloramphenicol, kanamycin, and tetracycline	Human urine	[82]
GCE	SGN-hematein/ILs/ penicillinase	DPV	Penicillin	0.0002	Levofloxacin hydrochloride, streptomycin sulfate, and Kanamycin sulfate,	Milk	[44,105]
GCE	Au/N-G	EIS and LSV	Chloramphenicol	591.0	Chlortetracycline, Oxytetracycline, and Metronidazole	Eye drop	[103]
GCE	Graphene	LSV	Midecamycin	101.0	Isovalerylspiramycin, acetylspiramycin, josamycin, and Kitasamycin	Urine and serum samples	[113]
GCE	Au/C_3_N_4_/GN	SWV	Ciprofloxacin and Chloramphenicol	421.0 and 28.0	-	Milk	[43]
GCE	Anti-Kan/WGS/PBCTS/NPG	SWV	Kanamycin	0.014	Neamine, neomycin, and gentamicin	Pork meat	[40]
GCE	β-cyclodextrin/rGO	DPV	Gatifloxacin	21.0	Norfloxacin, ofloxacin, ciprofloxacin, and moxifloxacin	Pharmaceuticals, and human urine	[112]
GCE	PoAP/GQD	DPV	Levofloxacin	11.0	Norfloxacin, lomefloxacin, enrofloxacin, and ciprofloxacin	Milk	[57]
GCE	Cl-RGO	DPV	Chloramphenicol	1000	Tetracycline, Erythromycin, penicillin G, and cysteine	Eye drops, water, calf plasma, and milk	[110]
GCE	CO_3_O_4_@rGO	Chronoamperometry and DPV	Chloramphenicol	551.0	glutathione, cysteine, and uric acid	Honey and milk	[108]
GCE	GO/ZnO	DPV	Chloramphenicol	11.0	4-amino phenol, 4-nitro phenol, 4-nitroaniline, 4-nitrobenzene, Cl^-^, and Ca^+2^	Eye drops, milk, and honey	[109]
GCE	PPy_3_C/ERGO	DPV	Streptomycin	0.6	Gentamycin, kanamycin, amikacin, neomycin, and dihydrostreptomycin	Honey and porcine kidney	[110]
GCE	3D RGO	DPV	Chloramphenicol	151.0	uric acid, cysteine, taurine, and glutathione	Milk and eye drops	[107]
GCE	MIP/G-AuNPs	DPV	Levofloxacin	531.0	Norfloxacin, prulifloxacin, oxytetracycline, and chlortetracycline	Levofloxacin capsule	[106]
GCE	Aptamer/HNP–PtCu/ GR-TH/GCE	DPV	Kanamycin	0.0009	Human chorionic gonadotropin, tyrosine, dopamine, TSH hormone.	Chicken liver and pork meat	[104]
GCE	Au-Pt Nps/MWCNT	LSV	Cefotaxime	1.0	Glucose, dopamine, and ascorbic acid	Plasma	[124]
Au	ssDNA/SWCNT	SWV	Levofloxacin	75.0	-	Urine	[117]
GCE	3DCNTs@ Cu NPs@MIP	CV	Chloramphenicol	10 × 10^3^	Florfenicol, clindamycin, Dansyl chloride, and thiamphenicol	Milk	[125]
Au	MIPs/GR-MWCNTs/ CS-SNP	Amperometry, and CV	Neomycin	7.65	Erythromycin, kanamycin, streptomycin, and gentamycin	Honey and milk	[123]
GCE	MWCNT-GNPs/MIP	CV	Tetracycline	91.0	Chloramphenicol, nafcillin, and oxytetracycline	-	[122]
Au	ssDNA/AuNPs/en/ MWCNTs	CV	Valrubicin	19.0	K^+^, paracetamol, Na^+^, glucose, urea, azithromycin, ascorbic acid, and Caffeine	Blood and human urine	[121]
GCE	MWCNT-CTAB-PDPA	Stripping DPV	Chloramphenicol	3.0	Streptomycin, ceftazidime, cefotaxime, and ceftizoxime	Honey and milk	[127]
Paraffin	MWCNT-Sb Nps	DPV	Trimethoprim, and Sulfamethoxazole	32.0 and 25.0	Carbaryl, and 17β Estradiol	Natural H_2_O	[126]
GCE	AgNPs/MWCNTsCOOH	DPV	Adriamycin	1.80	-	Ct-DNA	[120]
GCE	Anti-TET/GA/(CS-PBGR)2/MWCNTs-CS	DPV	Tetracycline	0.006	Gentamycin sulfate, kanamycinmonosulfate, and oxytetracycline	Milk	[119]
GCE	Anti-TET/MWCNTs	DPV	Tetracycline	6.0	Doxycycline hydrochloride and oxytetracycline	Milk	[118]
GCE	Si–Fe/NOMC/GCE	DPV	Chloramphenicol	31.0	Florfenicol, benzylpenicillin potassium, chlortetracycline hydrochloride, gentamicin sulfate, and thiamphenicol	Eye drop	[129]
GCE	CS @MnO_2_	DPV	Chlortetracycline	261.0	Rifampicin, Oxytetracycline, and chloramphenicol	Fish, shrimp, and milk	[130]
GCE	CB-DHP	SWV	Amoxicillin	121.0	Humic acid, vermicompost, albumin, glucose, K^+^, and Na^+^	Water and urine	[132]
SPE	CdS-KAP+ PbS-STP/ cKAP+ cSTP/OMCAuNPs/CNF	DPV	Streptomycin and kanamycin	0.05 and 0.09	Oxytetracycline, tobramycin, neomycin, and gentamycin	Milk	[35]
GCE	Aptamer/ SnOx@TiO_2_@mC	EIS	Tobramycin	0.02	Doxycycline, oxytetracycline, and Kanamycin	Human serum and urine	[133]
GCE	Aptamer/Fe_3_O_4_@mC	EIS	Oxytetracycline	0.00006	Chlortetracycline, doxycycline, and tetracycline	Milk	[134]
GCE	Nanodiamonds	SWV	Pyrazinamide	221.0	-	Human serum and urine	[135]
GCE	STR Aptamer/GRFe_3_O_4_-AuNPs/PCNR	DPV	Streptomycin	0.05	Glucose, methionine, ascorbic acid, and penicillin	Milk	[136]

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
