# Peer review of "The Application of Nanomaterials for the Electrochemical Detection of Antibiotics: A Review"

_micromachines, 2021, doi:10.3390/mi12030308_

Round 1
Reviewer 1 Report
The article reviews recent developments on the application nanomaterials in identification of antibiotics. In general, the paper is well-written, concise, comprehensive and to the point. However, it fails to give a general big picture of the significance on nanomaterials in antibiotics biosensing at the introduction. Although a detailed account of various techniques is given and these techniques are categorized, the reader is lost at the beginning. The reviewer suggests the authors to give a brief and general introduction on the various types of antibiotic biosensors, for example in form of a graphic illustration, before beginning the detailed discussions and categorizations.
Below are my comments:
Section 2. Electrochemical methods: Why voltammetric sensors first? It is desirable to briefly explain various voltammetry techniques, i.e. CV, DPV, and SWV before making comparisons in Table I. I would also suggest the discussion about the methods follow that of the detection strategies.
Section 2.1.2. Define “aptasensors”. What is the main difference between the functionality of aptamer-based biosensors made using the two different (immobilization of?) techniques mentioned? Also these two techniques are not explained clearly.
Section 2.2.1 QDs: line 161-162: How can a sensor have a recovery > 100%? Explain. What does ‘recovery’ refer to?
Section 2.2.3. The operation principle of magnetic NPs is not explained. Line 312: How does an electrochemical immunosensor combined with magnetic NPs work? In Fig. 1 define SPE, IL, & TET. Mention if the figure was reproduced or adapted from the reference. It is difficult to relate various voltammograms of Fig. 1b to the discussion in the text.
Section 2.2.4 Metal nanoparticles: Most of the sections including this one begin we exaggerative statements such as “ … nanoparticles are a class of martials with exceptional structure.” As a result, confusing the author to recognize which technic or material is eventually the best. Please avoid these kinds of statements unless a quantitative comparison is given. Also, I’d suggest moving this section to precede that of magnetic NPs.
Same comments for Fig. 1 applies to Fig. 2.
Line 703: “The excellent biocompatibility and conductibility of multi-walled carbon nanotubes…” The authors probably meant ‘conductivity’.
English needs some moderate correction.
Author Response
Reviewer 1:
Section 2.
Electrochemical methods: Why voltammetric sensors first?
The advantages of using the voltammetry technique were added.
It is desirable to briefly explain various voltammetry techniques, i.e. CV, DPV, and SWV before making comparisons in Table I.
Various techniques were briefly explained before Table 1.
I would also suggest the discussion about the methods follow that of the detection strategies.
Discussion added after detection strategies.
Section 2.1.2.
Define “aptasensors”.
Aptasensors were defined
What is the main difference between the functionality of aptamer-based biosensors made using the two different (immobilization of?) techniques mentioned? Also these two techniques are not explained clearly.
The difference between the two immobilization methods was added with explanations.
Section 2.2.1 QDs: line 161-162: How can a sensor have a recovery > 100%? Explain. What does ‘recovery’ refer to?
Recovery was defined and the discussion of how sensor has a recovery over 100%.
Section 2.2.3.
The operation principle of magnetic NPs is not explained.
The operation principle of magnetic NPs is explained.
Line 312: How does an electrochemical immunosensor combined with magnetic NPs work?
Magnetic NPs-based immunosensor fabrication was discussed with the role of each component in the sensing process.
In Fig. 1 define SPE, IL, & TET. Mention if the figure was reproduced or adapted from the reference.
SPE, IL, & TET meanings were added in the main text.
It is difficult to relate various voltammograms of Fig. 1b to the discussion in the text.
The figures were mentioned as reproduced. The a to h concentrations related to the voltammograms were added in the text.
Section 2.2.4
Metal nanoparticles: Most of the sections including this one begin we exaggerative statements such as “ … nanoparticles are a class of martials with exceptional structure.” As a result, confusing the author to recognize which technic or material is eventually the best. Please avoid these kinds of statements unless a quantitative comparison is given.
These words were deleted.
Also, I’d suggest moving this section to precede that of magnetic NPs.
This section (metal NPs) was moved before magnetic NPs.
Same comments for Fig. 1 applies to Fig. 2.
More discussions for Figure was added
Line 703: “The excellent biocompatibility and conductibility of multi-walled carbon nanotubes…” The authors probably meant ‘conductivity’.
We mean conductibility that related to the ability of material to conduct heat or electricity and used in many research articles as the ref. (Wang, Y.; Hu, S. Applications of carbon nanotubes and graphene for electrochemical sensing of environmental pollutants. Journal of Nanoscience and Nanotechnology 2016, 16, 7852-7872. https://doi.org/10.1166/jnn.2016.12762)
English needs some moderate correction
The manuscript has been thoroughly reviewed and the written errors were corrected.
Reviewer 2 Report
The authors have reviewed many articles for their review-manuscript. I have read this manuscript with extreme interest in the topic because antibiotic resistance and novel antibiotic analysis systems are growing year by year. It could be considered for publish after proper revision. The specific comments are as follows.
- Figure 1 and Figure 2 resolution and level should improve for study.
- More figures or schematic diagrams need to add that will increase attraction for the reader.
- The advantages and demerits of different detection methods need to elaborate more clearly.
- The authors could add antibiotic sensitivity and how such effects hamper the typical antibiotics analysis.
- The scope of the advancement of the nanomaterials-based antibiotic analysis systems should be mention in the conclusion section.
Author Response
Reviewer 2:
- Figure 1 and Figure 2 resolution and level should improve for study.
The figures added again with the improved resolution.
- More figures or schematic diagrams need to add that will increase attraction for the reader.
More figures added to the paper.
- The advantages and demerits of different detection methods need to elaborate more clearly.
The advantages and disadvantages of each type of nanomaterials were added to each section and the comparison between different methods was discussed in section 3.
- The authors could add antibiotic sensitivity and how such effects hamper the typical antibiotics analysis.
There is no relation between antibiotics sensitivity and their detection in different fluids and samples. If the author means the sensor sensitivity toward antibiotics (sensitivity supposed to add to table 2 but we omitted this column because the table is too long)
- The scope of the advancement of the nanomaterials-based antibiotic analysis systems should be mention in the conclusion section.
The scope of nanomaterials-based antibiotic sensors was mentioned in the conclusion section.
Round 2
Reviewer 1 Report
Some parts are corrected and new figures are added but it is still extremely confusing to follow the story of the new figures. For example in Fig. 4 has repeated panels.
Author Response
Some parts are corrected and new figures are added but it is still extremely confusing to follow the story of the new figures. For example in Fig. 4 has repeated panels.
This repetition of figures is due to the word tracking system that shows the old and new figures. So, we stopped the tracking option and highlighted all changes instead.
Reviewer 2 Report
The editor can consider accepting the review manuscript.
Author Response
The editor can consider accepting the review manuscript.
I think the reviewer become satisfied with the paper
Round 3
Reviewer 1 Report
The paper can now be published in Micromachines.